# TextME: Text-only Training for Modality Expansion via LLM Space Pivoting

## Abstract

Expanding multimodal representations to novel modalities is constrained by reliance on large-scale paired datasets (e.g., text–image, text–audio, text–3D, text–molecule), which are costly and often infeasible in domains requiring expert annotation such as medical imaging, 3D modeling, and molecular analysis. We introduce TextME, to our knowledge the first framework for text-only modality expansion that removes paired data requirements. Our method leverages the universal geometric properties of pre-trained encoders—consistent modality gaps—which enable zero-shot cross-modal transfer once embedding spaces satisfy these properties. We empirically verify that these hold across audio, 3D, X-ray, and molecular domains, enabling effective cross-modal tasks without paired supervision. Furthermore, we evaluated LLM and multimodal text encoders to determine which is more effective as a unified anchor space. Experiments show that TextME achieves 91.5% of paired-data performance in zero-shot classification and cross-modal retrieval, while also supporting emergent capabilities between unseen modality pairs (e.g., audio-to-3D, molecule-to-image). These results highlight text-only modality expansion as a practical and scalable path toward foundation models spanning arbitrary modalities.

## 1 Introduction

*Modality expansion*, which aligns heterogeneous data modalities into a unified embedding space, has emerged as a core challenge in multimodal representation learning (Baltrušaitis et al., 2018; Manzoor et al., 2023; Liang et al., 2024; Yuan et al., 2025; Liu et al., 2025). Recent approaches leverage large-scale paired datasets to project diverse modalities—such as images, audio, and 3D point clouds—into shared semantic spaces where equivalent content maintains proximity (Zhang et al., 2023a; Han et al., 2023; Zhu et al., 2023; Lyu et al., 2024; Wang et al., 2023a). While text–image and text–audio corpora have enabled remarkable progress in vision–language (Radford et al., 2021; Jia et al., 2021) and audio–language (Wu et al., 2023; Manco et al., 2022) modeling, the construction of such paired resources proves prohibitively expensive or infeasible for many specialized domains. Medical imaging requires costly expert annotations while navigating privacy constraints (Wang et al., 2025; Kitamura et al., 2024; Ziller et al., 2021), molecular analysis demands complex domain-specific representations (Edwards et al., 2024; Xiao et al., 2024), and 3D modeling necessitates labor-intensive curation (Deitke et al., 2023; Sarkar et al., 2025). Consequently, the scalability of modality expansion is constrained not merely by architectural limitations but, more fundamentally, by the availability of paired supervision.

Recent advances (Wang et al., 2023c; Zhang et al., 2024b; Wang et al., 2024a;b) demonstrate that pre-trained multimodal encoders like CLIP (Radford et al., 2021) and ALIGN (Jia et al., 2021) can be effectively reused through lightweight projection networks to integrate multiple modalities into shared representation spaces. However, these approaches still require fully-paired multimodal data during training, demanding simultaneous access to all target modalities with corresponding supervision. This requirement becomes particularly challenging when extending to modalities beyond standard vision-language pairs—such as audio, 3D point clouds, medical X-rays, and molecular structures—where natural correspondences are often absent and domain expertise is scarce.

In this work, we eliminate the paired data requirement by exploiting an inherent geometric property of pre-trained multimodal encoders—the consistent modality gap. Inspired by prior theoretical work

demonstrating this phenomenon (Liang et al., 2022; Zhang et al., 2023b), we propose TextME, a framework that leverages this gap for text-only training. Zhang et al. (2024a) demonstrated that when a directionally consistent *offset* vector exists between pre-trained image and text embedding spaces, cross-modal transfer can be achieved via simple vector translation without paired training. We extend this insight by empirically verifying that such modality gaps are a universal property of contrastively-trained encoders, regardless of the specific modality they encode. Since these encoders leverage text for alignment during training, TextME exploits their shared text embedding space as a semantic anchor, applying pre-computed *offset* translation to bridge modality spaces using only text descriptions. We further validate the effectiveness of different anchor spaces, comparing LLM embeddings and multimodal text encoders as semantic anchors based on their capacity to capture cross-domain semantic similarity.

Our work differs fundamentally from prior modality gap analyses (Liang et al., 2022; Zhang et al., 2024a; Levi & Gilboa, 2024) that focus on *identifying* and *correcting* gaps in paired settings. While these works require multimodal paired supervision, TextME exploits the gap property to achieve modality expansion with only a few samples per modality—a 95% data reduction compared to methods like LanguageBind (Zhu et al., 2023). Similarly, while LLM-hub approaches (Lyu et al., 2024; Huang et al., 2024; Han et al., 2023) use text as a semantic anchor, they still require paired multimodal data during training. To the best of our knowledge, TextME is the first to combine text-only training with modality expansion capabilities.

We validate TextME's effectiveness through comprehensive experiments on diverse modalities using zero-shot classification and cross-modal retrieval tasks. Despite training exclusively on text descriptions, TextME achieves an average of 91.5% performance preservation compared to paired-data methods, with specific tasks like molecular retrieval, even surpassing supervised learning baselines. Moreover, our framework enables emergent cross-modal capabilities between modality pairs that have never seen during training, such as audio-to-3D and molecule-to-image retrieval, demonstrating that text-anchored alignment creates meaningful semantic bridges across arbitrary modalities.

Our contribution is three-fold:

- We provide comprehensive empirical validation that the consistent modality gap—a systematic offset between text and non-text embeddings—exists universally across diverse pre-trained encoders (e.g., audio, 3D, X-ray, and molecule). We demonstrate that this gap operates orthogonally to semantic content, enabling zero-shot cross-modal transfer without requiring paired multimodal data.

- We propose TextME, the first framework that exploits this geometric consistency to achieve modality expansion using only text descriptions. By leveraging LLM embeddings as a unified semantic anchor, our method captures richer semantic relationships across diverse domains.

- We demonstrate that text-only training can achieve 91.5% of paired-data performance across diverse modalities (i.e., audio, 3D, X-ray, molecule) while eliminating the need for target modality data during training. TextME enables emergent cross-modal retrieval capabilities between modality pairs that were never seen during training (e.g., audio-to-3D, molecule-to-image retrieval), demonstrating that text serves as an effective semantic bridge across arbitrary modalities.

## 2 THEORETICAL FOUNDATION: CROSS-MODAL INSTANCE MAPPING

We establish the theoretical underpinnings for text-only modality expansion by analyzing the geometric structure of pre-trained multimodal encoders. Our investigation reveals that contrastively-trained encoders exhibit a consistent modality gap—a systematic offset between text and non-text embeddings—enabling zero-shot cross-modal transfer through simple offset translation.

### 2.1 GEOMETRIC PROPERTIES OF CROSS-MODAL ALIGNMENT

Building on observations of vision-language models (Liang et al., 2022; Zhang et al., 2023b; 2024a), we extend the theoretical analysis to diverse specialized modalities, including audio, 3D, medical imaging, and molecular structures. We identify three critical hypotheses that support offset-based

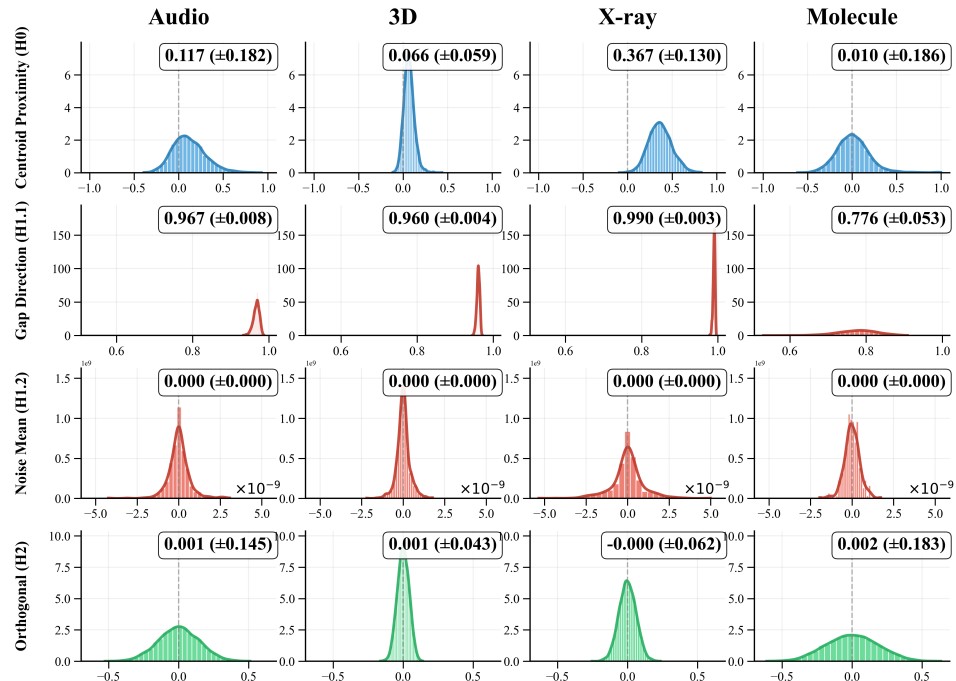

Figure 1: Geometric analysis of modality gaps across four multimodal encoders. **Centroid Proximity** (top): Mean pairwise cosine similarity from embeddings to modality centroids, where values near 0 indicate statistical independence. **Gap Direction** (second): Cosine similarity between instance-level offsets $\Delta_{ij}^{(k)}$ and group-level offset $\Delta_{ij}$, demonstrating consistency. **Noise Mean** (third): Distribution of per-dimension mean values of alignment noise (displayed as $\times 10^{-9}$). Each histogram shows the distribution of the $d$ components of $\bar{\epsilon} = E[\epsilon_k]$. **Gap Orthogonality** (bottom): Cosine similarity between gap vectors and semantic content variations within modalities.

alignment: intra-modal clustering (Hypothesis 0), inter-modal gap consistency (Hypothesis 1), and orthogonality between gap and content variations (Hypothesis 2).

**Definition 1** (Cross-Modal Instance Mapping). *Given a set of modalities $\mathcal{M} = \{m_1, m_2, \ldots, m_k\}$ with embeddings in a shared space $\mathbb{R}^d$, a cross-modal instance mapping is a transformation $\Phi_{ij} : \mathbb{R}^d \to \mathbb{R}^d$ that aligns embeddings from modality $m_i$ to modality $m_j$:*

$$\Phi_{ij}(e) = e - \Delta_{ij} \tag{1}$$

*where $\Delta_{ij} = \mu_{m_i} - \mu_{m_j}$ is the inter-modal offset between modality centroids $\mu_{m_i} = \mathbb{E}[e_{m_i}]$ and $\mu_{m_j} = \mathbb{E}[e_{m_j}]$. The mapping enables zero-shot cross-modal transfer when Hypotheses 0–2 are satisfied, ensuring that semantically corresponding embeddings $e_i, e_j$ satisfy $\|\Phi_{ij}(e_i) - e_j\| < \delta$ for small $\delta$.*

**Hypothesis 0** (Intra-Modal Alignment Independence). *For a modality $m \in \mathcal{M}$, normalized embeddings exhibit statistical independence from the modality centroid: embeddings $\{e_i\}$ from modality $m$ satisfy $\mathbb{E}[\cos(\hat{e}_i, \hat{\mu}_m)] \approx 0$, where $\hat{e} = e/\|e\|_2$ denotes normalization and $\hat{\mu}_m$ is the normalized centroid.*

Near-zero cosine similarity indicates that embeddings are orthogonal to the modality centroid, reflecting statistical independence rather than tight clustering. This independence property emerges from the $\ell_2$ normalization and contrastive objective, which distributes embeddings across the unit hypersphere. We empirically verify this through Centroid Proximity statistics, where near-zero values confirm statistical independence.

**Hypothesis 1** (Inter-Modal Gap Consistency). *For modalities $m_i, m_j \in \mathcal{M}$, a consistent offset exists between their embedding spaces:*

**Hypothesis 1.1** (Space-level Gap Consistency). *Instance-level offsets can be approximated by a single group-level offset: $\Delta_{ij}^{(k)} \approx \Delta_{ij}$ for all instance pairs $k$, where $\Delta_{ij} = \mu_i - \mu_j$ is the difference between modality centroids.*

The modality gap originates from inherent differences in modality characteristics and architectural properties from initializing separate encoders. We verify this through Gap Direction analysis, measuring an average cosine similarity between instance-level and group-level offsets to assess directional consistency across the dataset.

**Hypothesis 1.2** (Instance-level Gap Consistency). *Deviations from the mean offset follow a bounded distribution:* $\epsilon_k = \Delta_{ij}^{(k)} - \Delta_{ij} \sim \mathcal{N}(0, \sigma^2)$ *where* $\sigma < \gamma \cdot \tau$, *with* $\tau$ *being the temperature parameter and* $\gamma$ *a modality-specific constant. Here,* $\Delta_{ij}^{(k)} = e_{modal}^{(k)} - e_{text}^{(k)}$ *denotes the instance-level offset between the modal encoder* $E_m^{modal}$ *and text encoder* $E_m^{text}$ *for paired sample* $k$.

The bound $\sigma < \gamma \cdot \tau$ arises from the contrastive learning objective, where the temperature parameter $\tau$ controls the concentration of embeddings. Following Chen et al. (2020); Zhang et al. (2024a), embeddings trained with InfoNCE naturally exhibit bounded variance proportional to $\tau$. We empirically validate this relationship through per-dimension analysis of alignment noise, measuring the distribution of noise magnitudes across all embedding dimensions to confirm bounded, zero-mean variations. The modality-specific constant $\gamma$ accounts for architectural differences between encoders but remains bounded by the training objective. See Appendix C.2 for detailed statistical methodology.

**Hypothesis 2** (Orthogonality of Inter/Intra Variations). *The inter-modal offset* $\Delta_{ij}$ *is orthogonal to intra-modal semantic variations:* $\Delta_{ij} \perp r_m^{(p,q)}$ *for any instances* $p, q$ *within modality* $m \in \{m_i, m_j\}$, *where* $r_m^{(p,q)} = e_p - e_q$ *denotes the difference vector between embeddings.*

This orthogonality indicates that $\Delta_{ij}$ operates independently of semantic content within each modality. Since the offset is orthogonal to semantic variations $r_m^{(p,q)}$, applying the mapping $\Phi_{ij}(e) = e - \Delta_{ij}$ preserves relative distances and semantic relationships. We verify this through Gap Orthogonality analysis, measuring the cosine similarity between gap vectors and within-modality semantic differences to assess independence from content variations.

## 2.2 Theoretical Justification for Orthogonality (H2)

We provide theoretical justification that the InfoNCE (Chen et al., 2020) objective used to train all evaluated encoders intrinsically enforces the noise-semantic orthogonality assumption (Hypothesis 2). We decompose embeddings as $e_x = s + \epsilon_x$ and $e_t = s + \epsilon_t$, where $s \in \mathbb{R}^d$ represents shared semantic content and $\epsilon_x, \epsilon_t \in \mathbb{R}^d$ represent modality-specific noise. All evaluated encoders (i.e., CLIP, CLAP, Uni3D, etc.) are trained with the InfoNCE contrastive objective:

$$\mathcal{L}_{\text{InfoNCE}} = -\mathbb{E}_{(e_x, e_t) \sim p_{\text{pos}}} \left[ \log \frac{\exp(e_x^\top e_t / \tau)}{\exp(e_x^\top e_t / \tau) + \sum_{j=1}^{N-1} \exp(e_x^\top e_t^{(j)} / \tau)} \right] \quad (2)$$

where $(e_x, e_t)$ is a positive pair, $\{e_t^{(j)}\}_{j=1}^{N-1}$ are negative samples, $\tau$ is temperature, and $N$ is batch size.

**Theorem 1** (InfoNCE Enforces Noise-Semantic Orthogonality). *Minimizing* $\mathcal{L}_{InfoNCE}$ *with respect to encoder parameters naturally enforces noise-semantic orthogonality:* $\mathbb{E}[\epsilon_x^\top s] \to 0$ *and* $\mathbb{E}[\epsilon_t^\top s] \to 0$, *cross-modal noise independence:* $\mathbb{E}[\epsilon_x^\top \epsilon_t] \to 0$, *and semantic alignment:* $\mathbb{E}[\|s_x - s_t\|^2] \to 0$ *for matched pairs.*

For a positive pair, the similarity decomposes as $e_x^\top e_t = \|s\|^2 + s^\top \epsilon_t + \epsilon_x^\top s + \epsilon_x^\top \epsilon_t$. If the noise $\epsilon_x$ were correlated with semantics ($\mathbb{E}[\epsilon_x^\top s] \neq 0$), it would also correlate with negative samples' semantic content $s^{(j)}$, inflating negative similarities $e_x^\top e_t^{(j)}$ and reducing the margin between positive and negative pairs. This directly increases the loss. The optimization gradient therefore pushes $\epsilon_x$ to become orthogonal to all semantic directions, enforcing $\mathbb{E}[\epsilon_x^\top s] = 0$ at optimum. This demonstrates that orthogonality is not merely an empirical observation but a necessary consequence of the training objective itself. Full proof is provided in Appendix B.

## 2.3 EMPIRICAL MODALITY GAP VALIDATION ACROSS DIVERSE MODALITIES

To validate our theoretical framework, we analyzed five pre-trained multimodal encoders spanning diverse modalities: LanguageBind (Zhu et al., 2023) for vision, CLAP (Elizalde et al., 2023) for audio, Uni3D (Zhou et al., 2023) for 3D point clouds, CXR-CLIP (You et al., 2023) for medical X-rays, and MoleculeSTM (Liu et al., 2023) for molecular structures. For each encoder, we randomly sampled $N = 5,000$ text-modal pairs from their training domains to compute the inter-modal offset $\Delta_{ij} = \mathbb{E}[\mathcal{E}_{m_i}(x)] - \mathbb{E}[\mathcal{E}_{m_j}(t)]$ and analyze its geometric properties.

Figure 1 presents comprehensive validation results demonstrating that all three hypotheses hold across diverse modalities. **Centroid Proximity** (top row) measures cosine similarity between individual embeddings and modality centroids to assess Hypothesis 0. 3D ($0.066 \pm 0.059$) and Molecule ($0.010 \pm 0.186$) exhibit near-zero values, indicating strong statistical independence, while Audio ($0.117 \pm 0.182$) and X-ray ($0.367 \pm 0.130$) show higher values with varying degrees of deviation from the independence assumption. The impact of these variations on downstream task performance is examined in Section 4.2. **Gap Direction** (second row) demonstrates $\cos(\Delta_{ij}^{(k)}, \Delta_{ij}) > 0.96$ consistency, validating single-vector characterization across modalities (Hypothesis 1.1). **Noise Mean** (third row) confirms zero-centered distributions with $\mathbb{E}[\epsilon_k] \approx 0$, indicating predictable alignment variations (Hypothesis 1.2). Following the per-dimension analysis methodology of Zhang et al. (2024a), we verify that this property holds not just globally but across all $d$ embedding dimensions independently, with values consistently below $10^{-8}$. This uniform zero-mean property across dimensions validates that the constant offset approximation does not introduce systematic bias in any direction of the embedding space. **Gap Orthogonality** (bottom row) shows $|\cos(\Delta_{ij}, r_m^{(p,q)})| < 0.05$, confirming that modality gaps operate independently of semantic content, enabling effective cross-modal transfer through simple offset operations (Hypothesis 2). These geometric properties establish a unified framework for understanding and exploiting cross-modal relationships in pre-trained encoders.

## 3 TEXT-ANCHORED MODALITY EXPANSION FRAMEWORK

Building on the theoretical insights from Section 2, we propose TextME, a framework that exploits the consistent modality gap property to enable text-only training for modality expansion. Our approach leverages the geometric consistency demonstrated in Section 2.1—that pre-trained encoders exhibit a constant offset between text and non-text embeddings orthogonal to semantic content. This property allows us to create an interchangeable coordinate system through simple centering operations, eliminating the need for paired multimodal data.

### 3.1 PROBLEM FORMULATION

Modality expansion aims to integrate diverse pre-trained encoders into a unified semantic space where similar concepts maintain proximity regardless of their originating modality. Consider a set of pre-trained encoders $\{E_m : \mathcal{X}_m \to \mathbb{R}^{d_m}\}$, where each encoder $E_m$ maps inputs from modality $m$'s input space $\mathcal{X}_m$ to $d_m$-dimensional embeddings. Given a source modality $m_s$ with established semantic representations and target modalities $\mathcal{M}_T = \{m_1, \ldots, m_k\}$ to be incorporated, the objective is to learn projection networks $P_m : \mathbb{R}^{d_m} \to \mathbb{R}^{d_h}$ that preserve semantic relationships across modalities, where $d_h$ denotes the dimensionality of the shared embedding space.

We consider a practical scenario where only unpaired textual descriptions $\mathcal{D}_{\text{text}} = \{t_i\}_{i=1}^N$ are available for training. Pre-trained multimodal models consist of text encoders $E_m^{\text{text}}$ and modal encoders $E_m^{\text{modal}}$ jointly optimized through contrastive learning. Our approach exploits the geometric properties identified in Section 2.1—specifically, the consistent offset between these encoders—to enable alignment without paired multimodal data.

### 3.2 FRAMEWORK OVERVIEW

TextME operates through three stages that decouple geometric alignment from semantic projection. First, we pre-compute modality-specific offsets $\Delta_m = \mu_m^{\text{modal}} - \mu_m^{\text{text}}$ using Equation 1 to characterize the geometric transformation between text and modal encoders. Second, we train lightweight projec-

tion networks $P_m$ exclusively on centered text embeddings, mapping them to a shared representation space using only unpaired text descriptions. Third, at inference, we apply the pre-computed offset to non-text modality embeddings, then project them using the text-trained network. This design exploits the orthogonality property (Hypothesis 1) to preserve semantic relationships while enabling cross-modal transfer without paired supervision. Algorithm 1 formalizes the complete procedure.

### 3.2.1 OFFSET COMPUTATION

We establish interchangeability between text and modal embedding spaces by pre-computing modality-specific offsets. For each encoder $E_m$, we compute centroids $\mu_m^{\text{text}} = \mathbb{E}[E_m^{\text{text}}(t)]$ and $\mu_m^{\text{modal}} = \mathbb{E}[E_m^{\text{modal}}(x)]$ over representative samples from each distribution. By centering each modality independently—subtracting $\mu_m^{\text{text}}$ from text embeddings and $\mu_m^{\text{modal}}$ from modal embeddings—we create a shared coordinate system where both modalities are aligned at the origin. This enables projection networks trained on centered text embeddings to generalize to centered modal embeddings at inference. The offset computation requires only 5,000 samples—a 99% reduction from typical paired training requirements (Zhu et al., 2023; Zhang et al., 2024b).

### 3.2.2 TEXT-TO-TEXT ALIGNMENT

**Contrastive Learning for Projection Network.** Given text descriptions $\mathcal{D}_{\text{text}} = \{t_i\}_{i=1}^N$ from the target modality domain, we train a lightweight projection networks $P_m : \mathbb{R}^{d_m} \to \mathbb{R}^{d_h}$ to map centered text embeddings into a shared representation space. Each projection network consists of a 2-layer MLP with GeLU activation, requiring only ~10M parameters. The training objective employs contrastive learning with hard negative mining (Lee et al., 2024; Moreira et al., 2024; Rösch et al., 2024):

$$\mathcal{L}_{\text{align}} = -\frac{1}{B} \sum_{i=1}^B \log \frac{\exp(\text{sim}(z_i, z_i')/\tau)}{\sum_{j \in \mathcal{N}_i \cup \{i\}} \exp(\text{sim}(z_i, z_j')/\tau)} \tag{3}$$

where $z_i = P_m(E_m^{\text{text}}(t_i) - \mu_m^{\text{text}})$ is the projected centered embedding, $z_i' = E_s(t_i)$ is the shared space embedding, and $\mathcal{N}_i$ contains hard negatives with similarity scores in $[0.1 \cdot \text{sim}(z_i, z_i'), 0.9 \cdot \text{sim}(z_i, z_i')]$. This sampling strategy accelerates convergence by focusing gradients on informative examples near the decision boundary.

**Choice of Shared Anchor Space.** We empirically validate two candidate text representation spaces as semantic anchors: LLM embeddings (i.e., NV-Embed-v2 (Lee et al., 2024) and Qwen3-Embeddings (Zhang et al., 2025)) and multimodal text encoders (i.e., Language-Bind (Zhu et al., 2023)). To assess whether the embeddings faithfully capture semantic representations, we evaluated their performance on the STS benchmark, a widely used metric for contextual understanding that measures sentence similarity. LLM embeddings achieve $85.79 \sim 90.40$ Spearman correlation on STS benchmarks versus $68.29 \sim 68.83$ for multimodal encoders (Table 4 in Appendix D), reflecting their superior semantic understanding from extensive next-token prediction training.

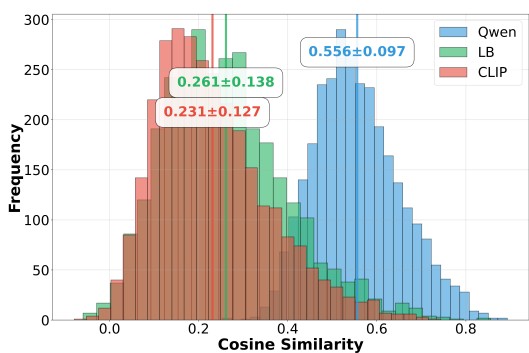

Figure 2: Semantic anchoring comparison between LLM embeddings and multimodal encoders on 3,000 semantically equivalent cross-modal description pairs.

To assess cross-domain alignment capabilities, we analyzed $3,000$ audio-image caption pairs from FlickrNet (Senocak et al., 2018), where we generated linguistically distinct but semantically equivalent descriptions using the Gemini API (Google, 2024). For instance, an image caption "a red sports car speeding on highway" was paired with its audio equivalent "loud engine roar with wind rushing past", testing whether encoders can recognize semantic similarity despite different surface forms. Figure 2 shows LLM embeddings (Qwen) exhibit clearer separation between matched and unmatched pairs (0.56 vs 0.23-0.26 mean cosine similarity for matched pairs) compared to multimodal encoders. This demonstrates their superior capability to discriminate semantically equivalent pairs from unrelated ones, validating their effectiveness as semantic anchors for text-only training.

We note that the downstream alignment quality is ultimately validated through retrieval and classification performance in Tables 1-2.

### 3.2.3 INFERENCE-TIME CROSS-MODAL TRANSFER

At inference, TextME enables zero-shot cross-modal capabilities through offset-based transformation. For a non-text input $x$ from modality $m$, we compute:

$$e_{\text{final}} = P_m(E_m^{\text{modal}}(x) - \mu_m^{\text{modal}}) \tag{4}$$

The centering operation $(E_m^{\text{modal}}(x) - \mu_m^{\text{modal}})$ transforms the modal embedding into the coordinate system used during text training. Since the offset is orthogonal to semantic variations (Hypothesis 2), this transformation preserves semantic relationships while enabling the text-trained projection network $P_m$ to map modal embeddings to the shared representation space, achieving effective cross-modal retrieval and classification without paired supervision.

## 4 EXPERIMENTS

We evaluate TextME's ability to expand multimodal representations through comprehensive experiments across diverse modalities. Our analysis includes quantitative evaluation on the standard benchmarks for cross-modal retrieval and zero-shot classification (Section 4.2), and qualitative examination of emergent cross-modal capabilities between modality pairs never paired during training (Section 4.3).

### 4.1 EXPERIMENTAL SETUP

**Source and Target Modalities.** We conducted experiments to verify the modality expansion capability of TextME. For the source representation space, we select LanguageBind (Zhu et al., 2023), an image-text aligned standard multimodal foundation model. As target modalities, we integrate four specialized domains that lack natural multimodal correspondences: **Audio** using CLAP (Elizalde et al., 2023) trained on AudioCaps (Kim et al., 2019) descriptions, **3D** using Uni3D (Zhou et al., 2023) trained on Cap3D-Objaverse (Luo et al., 2023) captions, **X-ray** using CXR-CLIP (You et al., 2023) trained on CheXpert (Irvin et al., 2019) reports, and **Molecule** using MoleculeSTM (Liu et al., 2023) trained on PubChem (Kim et al., 2025) descriptions. For fair comparisons across modalities, we sample $100K$ text descriptions from each modality-specific training dataset.

**Text Anchor Space.** To establish a unified text embedding space, we utilize three distinct models: LanguageBind (LB; Zhu et al. 2023), NV-Embed-v2 (NV; Lee et al. 2024), and Qwen3-Embedding-4B (Qwen; Zhang et al. 2025). These models are carefully selected to evaluate the efficacy of our proposed framework across diverse shared anchor spaces, each exhibiting different representational capabilities as Section 3.2.2. For clarity, we denote the corresponding implementations as **Ours**$_{LB}$, **Ours**$_{NV}$, and **Ours**$_{Qwen}$.

**Baselines.** We compare against two categories of methods to evaluate TextME's effectiveness. *Paired*-data methods include LanguageBind (Zhu et al., 2023), which trains modality-specific encoders from scratch, and Ex-MCR (Zhang et al., 2024b), which adapts frozen pre-trained encoders. For direct comparisons, we implement **Ours**$_{upper\text{-}bound}$, a variant of TextME with the same architecture but trained on paired multimodal data, representing the performance upper bound. *Unpaired*-data methods lack established baselines for our zero-shot setting. We therefore compare our approach with COX (Huang et al., 2025), which fine-tunes target modalities without instance-level pairing, although it requires labeled target data, unlike our approach. We re-implemented COX following the specification of the paper; details are given in Appendix G.2. We also include a Naïve baseline using PCA projection to the source embedding space (i.e., 768 dimensions for LanguageBind) with standard normalization, demonstrating that simple dimensionality reduction without learned alignment is insufficient.

**Evaluation Tasks.** To verify the effectiveness of TextME, we evaluate its performance on two categories of cross-modal downstream tasks. For **cross-modal retrieval**, we evaluate: (i) *Text→X*

Table 1: Zero-shot cross-modal retrieval performance. Highlighted rows share identical architecture but differ only in training data type (paired multimodal vs. text-only) and LLM anchoring. *Avg. Preservation* represents the average percentage of the supervised upper bound ($\mathbf{Ours}_{upper\text{-}bound}$) achieved by each TextMEvariant, computed across R@1 and R@5 metrics. † indicates our reproduction due to unavailable public code. **Bold** indicates best among unsupervised methods.

| | Text → Audio | | | | Text → Molecule | | Audio → Image | |
| | AudioCaps | | Clotho | | DrugBank | | FlickrNet | |
| Method | R@1 | R@5 | R@1 | R@5 | MRR@10 | MRR@20 | R@1 | R@5 |
|---|---|---|---|---|---|---|---|---|
| *Paired* | | | | | | | | |
| LanguageBind | 12.42 | 36.70 | 11.32 | 31.03 | – | – | 1.52 | 6.36 |
| Ex-MCR | 19.07 | 47.05 | 7.01 | 22.04 | – | – | 1.57 | 5.94 |
| $\mathbf{Ours}_{upper\text{-}bound}$ | 19.79 | 51.48 | 9.53 | 26.56 | 27.97 | 22.03 | – | – |
| *Unpaired* | | | | | | | | |
| Naïve | 0.02 | 0.35 | 0.04 | 0.23 | 10.17 | 4.24 | 0.02 | 0.06 |
| COX† | 0.08 | 0.64 | 0.11 | 0.78 | 7.63 | 2.54 | 0.02 | 0.10 |
| $\mathbf{Ours}_{CLIP}$ | 15.91 | 42.06 | 6.60 | 22.66 | – | – | 0.82 | **3.52** |
| $\mathbf{Ours}_{LB}$ | 14.54 | 41.02 | 6.93 | 22.33 | 29.66 | 20.34 | 0.92 | 3.42 |
| $\mathbf{Ours}_{NV}$ | **16.20** | **45.15** | 7.75 | 23.73 | 26.27 | 22.88 | 0.74 | 3.28 |
| $\mathbf{Ours}_{Qwen}$ | 15.35 | 43.88 | **7.81** | **23.81** | **31.36** | **26.27** | **1.06** | 3.14 |
| *Avg. Preservation* | 78.3% | 83.6% | 76.3% | 87.1% | 104.0% | 105.1% | – | – |

Table 2: Zero-shot classification performance across diverse modalities.

| | Audio | | | 3D | | | | X-ray |
| | AudioSet | ESC-50 | | ModelNet40 | | ScanObjectNN | | RSNA |
| Method | mAP | Top-1 | Top-5 | Top-1 | Top-5 | Top-1 | Top-5 | Top-1 |
|---|---|---|---|---|---|---|---|---|
| *Paired* | | | | | | | | |
| LanguageBind | 18.33 | 94.00 | 99.70 | – | – | – | – | – |
| Ex-MCR | 6.67 | 71.20 | 96.80 | 66.53 | 93.60 | 40.31 | 77.20 | – |
| $\mathbf{Ours}_{upper\text{-}bound}$ | 6.67 | 70.55 | 94.25 | 81.85 | 97.00 | 61.56 | 88.44 | 52.71 |
| *Unpaired* | | | | | | | | |
| Naïve | 1.14 | 2.90 | 8.45 | 0.81 | 8.95 | 3.32 | 30.52 | 26.36 |
| COX† | 1.26 | 2.00 | 10.00 | 4.05 | 13.70 | 2.84 | 26.68 | **72.38** |
| $\mathbf{Ours}_{CLIP}$ | 5.99 | **86.70** | **98.25** | 78.04 | 95.54 | **54.98** | **85.05** | 48.31 |
| $\mathbf{Ours}_{LB}$ | **6.42** | 74.65 | 94.60 | **81.12** | **97.49** | 54.81 | 84.88 | 49.04 |
| $\mathbf{Ours}_{NV}$ | 5.13 | 79.40 | 97.20 | 76.30 | 94.37 | 40.24 | 75.95 | 37.81 |
| $\mathbf{Ours}_{Qwen}$ | 5.80 | 77.25 | 96.85 | 70.86 | 92.14 | 42.15 | 77.89 | 39.61 |
| *Avg. Preservation* | 87.5% | 112.7% | 102.6% | 93.6% | 97.8% | 78.1% | 91.5% | 82.9% |

retrieval on AudioCaps (Kim et al., 2019), Clotho (Drossos et al., 2020), and DrugBank (Knox et al., 2024) (using MRR@k for molecules following MoleculeSTM (Liu et al., 2023)); (ii) $X \rightarrow X$ retrieval on Flickr30k (Plummer et al., 2015) for Audio→Image, demonstrating emergent cross-modal capabilities between modalities never paired during training. For **zero-shot classification**, we test on ModelNet40 (Qiu et al., 2021) and ScanObjectNN (Uy et al., 2019) for 3D point clouds, AudioSet (Gemmeke et al., 2017) and ESC-50 (Piczak) for audio, and RSNA Pneumonia Detection (RSNA, 2018) for X-ray images, reporting top-$k$ accuracy and mAP.

## 4.2 ZERO-SHOT CROSS-MODAL TASK PERFORMANCE

Tables 1 and 2 demonstrate that TextME achieves competitive zero-shot performance across diverse modalities. On cross-modal retrieval tasks (Table 1), TextME achieves 76.3%–87.1% preservation on Audio retrieval and exceeds paired-data performance on Molecule retrieval (104.0%–105.1% on DrugBank). On zero-shot classification tasks (Table 2), our method achieves 93.6%–97.8% preservation on ModelNet40 (3D) and 78.1%–91.5% on ScanObjectNN, while Audio and X-ray show

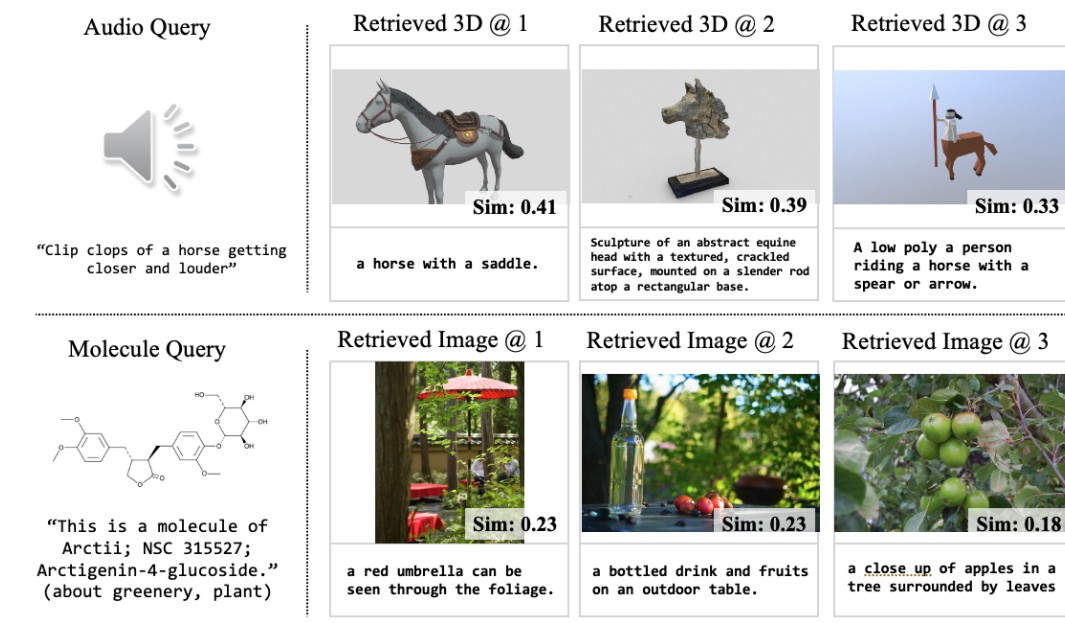

Figure 3: Emergent cross-modal retrieval without paired supervision. Audio queries retrieve semantically related 3D objects (top), and molecular structures retrieve contextually appropriate images (bottom). Results obtained by sampling 5,000 instances per modality and performing cosine similarity-based retrieval in the learned embedding space.

more variable performance across different benchmarks. This variability correlates with Hypothesis 0 satisfaction: modalities with lower centroid proximity—Molecule (0.010) and 3D (0.066)—achieve consistently high preservation, while Audio (0.117) and X-ray (0.367) with higher values show greater performance fluctuation across tasks. Overall, TextME achieves an average of 91.5% performance preservation compared to paired-data methods while significantly outperforming unpaired baselines such as COX and the naïve projection, validating the effectiveness of leveraging geometric consistency in pre-trained encoders.

Furthermore, our evaluation across different anchor spaces—multimodal encoders (Language-Bind (Zhu et al., 2023)) versus LLMs (NV-Embed-v2 (Lee et al., 2024), Qwen3-Embedding-4B (Zhang et al., 2025))—reveals task-dependent performance patterns: LLM anchors pretrained on diverse text corpora excel at retrieval through richer semantic relationships, while multimodal anchors pretrained on vision-language pairs develop discriminative boundaries better suited for classification. A more rigorous analysis of how pretraining objectives and data distributions influence anchor space properties remains valuable future work.

### 4.3 Qualitative Analysis of Cross-modal Capabilities

To evaluate emergent cross-modal transfer, we conducted retrieval experiments between modality pairs absent from training data, sampling 5,000 instances each from AudioCaps (Kim et al., 2019), Objaverse (Deitke et al., 2023), PubChem (Kim et al., 2025), and COCO (Lin et al., 2014). Results in Figure 3 demonstrate semantic coherence across modalities: audio-to-3D retrieval correctly associates equestrian sounds with horse models, and molecule-to-image retrieval matches pharmaceutical compounds with semantically related visual scenes. These observations confirm that text-only training with offset correction preserves semantic information from non-text modalities during inference.

## 5 RELATED WORK

Table 3 summarizes the key distinctions between TextME and prior approaches in modality gap analysis, LLM-hub methods, and modality expansion.

Table 3: Comparison of TextME with related work. Unlike prior modality gap methods that require paired data for gap correction, and LLM-hub approaches that need large-scale multimodal supervision, TextME uniquely combines gap exploitation with text-only training for modality expansion.

| Category | Method | Training Data | Gap Utilization | Modality Expansion |
|----------|--------|---------------|-----------------|--------------------|
| **Gap Analysis** | Mind the Gap (Liang et al., 2022) | Paired | Discovery only | ✗ |
| | $C^3$ (Zhang et al., 2024a) | Paired | Zero-centering | ✗ |
| | I0T (An et al., 2025) | Paired | Standardization | ✗ |
| **LLM Hub** | UniBind (Lyu et al., 2024) | Paired | ✗ | ✓ |
| | LLM2CLIP (Huang et al., 2024) | Text-only | ✗ | ✗ |
| | E5-V (Jiang et al., 2024) | Text-only | ✗ | ✗ |
| **Modality Expansion** | LanguageBind (Zhu et al., 2023) | Paired | ✗ | ✓ |
| | Ex-MCR (Zhang et al., 2024b) | Paired | ✗ | ✓ |
| | **TextME (Ours)** | **Text-only** | **Zero-centering** | ✓ |

**Modality Gap Analysis.** The modality gap—a systematic offset between text and non-text embeddings in contrastive models—has been studied from three perspectives. Liang et al. (2022) first identified this phenomenon in CLIP, Levi & Gilboa (2024) analyzed its geometric structure, and Zhang et al. (2024a) provided comprehensive statistical validation with zero-centering methodology. Recent work focuses on reducing gaps through linear separability (Shi et al., 2023), learnable correction models (Park et al., 2024; Eslami & de Melo, 2024), zero-centering for search (Li et al., 2025), and embedding standardization (An et al., 2025). While prior work discovers or corrects gaps in paired-data settings, TextME is the first to systematically exploit these geometric properties to enable modality expansion using only text descriptions.

**LLM Hub Approaches.** Recent work leverages LLMs as semantic anchors for multimodal alignment. Generative approaches include Han et al. (2023) for multi-modality instruction tuning, Han et al. (2024) for unified generation across modalities, and Xiao et al. (2025) for language-centric omnimodal learning through caption decoders. Representation enhancement methods include Lyu et al. (2024) using LLM-augmented unified spaces, Jiang et al. (2024) aggregating multimodal LLM features with frozen encoders, and Huang et al. (2024) leveraging LLMs for dense caption understanding. However, all these approaches require paired multimodal data during training—for instance, UniBind (Lyu et al., 2024) and LLM2CLIP (Huang et al., 2024) train on 60M+ image-caption pairs. TextME is the first to combine LLM-anchored alignment with text-only training, enabling modality expansion to specialized domains that lack natural paired data.

**Modality Expansion.** Contrastive learning enables multimodal alignment (Radford et al., 2021; Jia et al., 2021), with extensions to multiple modalities via central hubs: ImageBind (Girdhar et al., 2023) uses images, while LanguageBind (Zhu et al., 2023) leverages text as semantic pivots. To reduce computational costs, recent methods connect frozen pre-trained encoders through lightweight projectors—C-MCR (Wang et al., 2023c) and Ex-MCR (Zhang et al., 2024b) learn adapters between encoders, while FreeBind (Wang et al., 2024a) and OmniBind (Wang et al., 2024b) ensemble multiple encoders per modality. However, all require instance-level paired supervision, which becomes prohibitive in specialized domains (e.g., medical imaging, molecular analysis) where paired data is scarce or infeasible. TextME eliminates this requirement through geometric gap exploitation.

## 6 CONCLUSION

We presented TextME, a text-only training framework leveraging the consistent modality gap in pre-trained encoders to enable zero-shot cross-modal transfer. Across audio, 3D, medical X-ray, and molecular domains, TextME achieved 91.5% of paired-data performance while eliminating paired multimodal supervision, demonstrating that text descriptions can effectively bridge arbitrary modalities. Our framework addresses the critical bottleneck of paired dataset scarcity in specialized domains, establishing a scalable paradigm for multimodal representation learning in resource-constrained settings. Future work will explore how modality-specific characteristics influence alignment quality and develop adaptive strategies for domains where geometric assumptions may not fully hold.

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

# A  GENERATIVE AI USAGE DISCLOSURE

During the preparation of this manuscript, the following generative AI tools were used:

- **GitHub Copilot** was used solely for code completion and code snippet generation during the development of experimental pipelines and auxiliary scripts. All generated code was manually reviewed and, where necessary, modified by the authors.
- **Grammarly, Perplexity and ChatGPT** were used only for grammar checking, typo correction, and minor language editing of author-written text. No sections of the paper were written or generated entirely by generative AI models; all scientific content, analysis, and claims were produced by the authors.

No generative AI tool was used to produce any scientific content, experimental results, or substantive text in the manuscript. The use of generative AI tools was strictly limited to the above purposes, in accordance with ICLR 2026 policy.

# B  THEORETICAL PROOFS

## B.1  PROOF: INFONCE ENFORCES NOISE-SEMANTIC ORTHOGONALITY

A potential concern is whether modality-specific noise $\epsilon$ can be correlated with semantic content, i.e., $\mathbb{E}[\epsilon_x^\top s] \neq 0$. Such correlation would violate the orthogonality assumption in Hypothesis 2. We prove that the InfoNCE contrastive objective intrinsically prevents this correlation, thereby validating our hypothesis.

Following the decomposition in Section 2.2, we denote $e_x \in \mathbb{R}^d$ as the embedding from modality $x$ (e.g., image, audio, 3D), $e_t \in \mathbb{R}^d$ as the text embedding, $s \in \mathbb{R}^d$ as the shared semantic component, and $\epsilon_x, \epsilon_t \in \mathbb{R}^d$ as modality-specific noise. The embeddings decompose as:

$$e_x = s + \epsilon_x, \quad e_t = s + \epsilon_t \tag{5}$$

All evaluated encoders are trained with the InfoNCE (Chen et al., 2020) objective:

$$\mathcal{L}_{\text{InfoNCE}} = -\mathbb{E}_{(e_x, e_t) \sim p_{\text{pos}}} \left[ \log \frac{\exp(e_x^\top e_t / \tau)}{\exp(e_x^\top e_t / \tau) + \sum_{j=1}^{N-1} \exp(e_x^\top e_t^{(j)} / \tau)} \right] \tag{6}$$

where $(e_x, e_t)$ is a positive pair, $\{e_t^{(j)}\}_{j=1}^{N-1}$ are negatives from different concepts, $\tau$ is temperature, and $N$ is batch size.

**Lemma 1.** *Under InfoNCE optimization, the optimal embedding structure satisfies* $\mathbb{E}[\epsilon_x^\top s] = 0$.

*Proof.* We proceed by contradiction. For a positive pair $(e_x, e_t)$ sharing semantic content $s$:

$$e_x^\top e_t = (s + \epsilon_x)^\top (s + \epsilon_t)$$
$$= \|s\|^2 + \underbrace{s^\top \epsilon_t}_{\text{text noise-semantic}} + \underbrace{\epsilon_x^\top s}_{\text{modal noise-semantic}} + \underbrace{\epsilon_x^\top \epsilon_t}_{\text{cross-noise}} \tag{7}$$

For a negative pair $(e_x, e_t^{(j)})$ where $e_t^{(j)} = s^{(j)} + \epsilon_t^{(j)}$ with $s^{(j)} \neq s$:

$$e_x^\top e_t^{(j)} = (s + \epsilon_x)^\top (s^{(j)} + \epsilon_t^{(j)})$$
$$= s^\top s^{(j)} + s^\top \epsilon_t^{(j)} + \epsilon_x^\top s^{(j)} + \epsilon_x^\top \epsilon_t^{(j)} \tag{8}$$

For randomly sampled negatives from different semantic concepts, we have:

$$\mathbb{E}_{s^{(j)} \neq s}[s^\top s^{(j)}] \approx 0 \tag{9}$$

This property is well-established in contrastive learning (Zhang et al., 2024a): different semantic concepts occupy approximately orthogonal directions in high-dimensional embedding spaces ($d \gg N$).

At the optimum of $\mathcal{L}_{\text{InfoNCE}}$, the positive pair must dominate all negatives:

$$e_x^\top e_t \gg e_x^\top e_t^{(j)} \quad \forall j \tag{10}$$

Now suppose $\mathbb{E}[\epsilon_x^\top s] = c \neq 0$ at some local optimum. For positive pairs:

$$\mathbb{E}[e_x^\top e_t] = \mathbb{E}[\|s\|^2] + c + \text{(other terms)} \tag{11}$$

For negative pairs, using Equation equation 9:

$$\mathbb{E}[e_x^\top e_t^{(j)}] = \mathbb{E}[\epsilon_x^\top s^{(j)}] + \text{(cross-noise terms)} \tag{12}$$

If $\epsilon_x$ correlates with semantic direction $s$ ($c \neq 0$), it must also have non-negligible projections onto other semantic directions $\{s^{(j)}\}$ in high-dimensional space. This inflates the negative similarities $e_x^\top e_t^{(j)}$, reducing the margin required by Equation equation 10 and thereby increasing $\mathcal{L}_{\text{InfoNCE}}$, contradicting the optimality assumption.

This can be seen explicitly from the gradient:

$$\frac{\partial \mathcal{L}_{\text{InfoNCE}}}{\partial \epsilon_x} \propto -s + \sum_{j=1}^{N-1} p_j \cdot s^{(j)} \tag{13}$$

where $p_j = \frac{\exp(e_x^\top e_t^{(j)}/\tau)}{\sum_k \exp(e_x^\top e_t^{(k)}/\tau)}$ are softmax weights. Since $\sum_j p_j \cdot s^{(j)}$ averages over multiple approximately orthogonal semantic directions, any persistent correlation $\epsilon_x^\top s \neq 0$ creates a non-zero gradient component that drives the optimization toward orthogonality.

Therefore, at the InfoNCE optimum, we must have $\mathbb{E}[\epsilon_x^\top s] = 0$ and $\mathbb{E}[\epsilon_t^\top s] = 0$, validating Hypothesis 2. $\square$

## C  STATISTICS OF MODALITY GAP

To analyze the geometric structure of pre-trained multimodal encoders, we compute several statistics that characterize the separation between text and non-text embedding spaces, following the methodology of Zhang et al. (2024a).

### C.1  STATISTICAL DEFINITIONS

For each encoder $E_m$, we randomly sample $N = 5000$ text-modal pairs $(t_i, x_i)$ and compute:

- **Individual gap**: $\mathbf{d}_j^{(i)} = \mathbf{e}_{x_j}^{(i)} - \mathbf{e}_{t_j}^{(i)}$ — the vector difference between paired embeddings.

- **Group gap**: $\mathbf{d}^{(i)} = \mathbb{E}_j[\mathbf{d}_j^{(i)}] = \frac{1}{N} \sum_{j=1}^N \mathbf{d}_j^{(i)}$ — the average gap across all pairs.

- **Gap length**: $\|\mathbf{d}^{(i)}\|_2$ — the magnitude of the average gap vector.

- **Gap direction consistency**: $\cos(\mathbf{d}^{(i)}, \mathbf{d}^{(j)}) = \frac{\mathbf{d}^{(i)} \cdot \mathbf{d}^{(j)}}{\|\mathbf{d}^{(i)}\|_2 \|\mathbf{d}^{(j)}\|_2}$ — cosine similarity between gap vectors from different sample sets.

- **Content orthogonality**: $\cos(\mathbf{d}^{(i)}, \mathbf{r}_{j,k}^{(i)})$ where $\mathbf{r}_{j,k}^{(i)} = \mathbf{e}_{x_j}^{(i)} - \mathbf{e}_{x_k}^{(i)}$ — cosine between gap and content variations.

- **Alignment noise**: $\epsilon_j^{(i)} = \mathbf{d}_j^{(i)} - \mathbf{d}^{(i)}$ — deviation from the average gap.

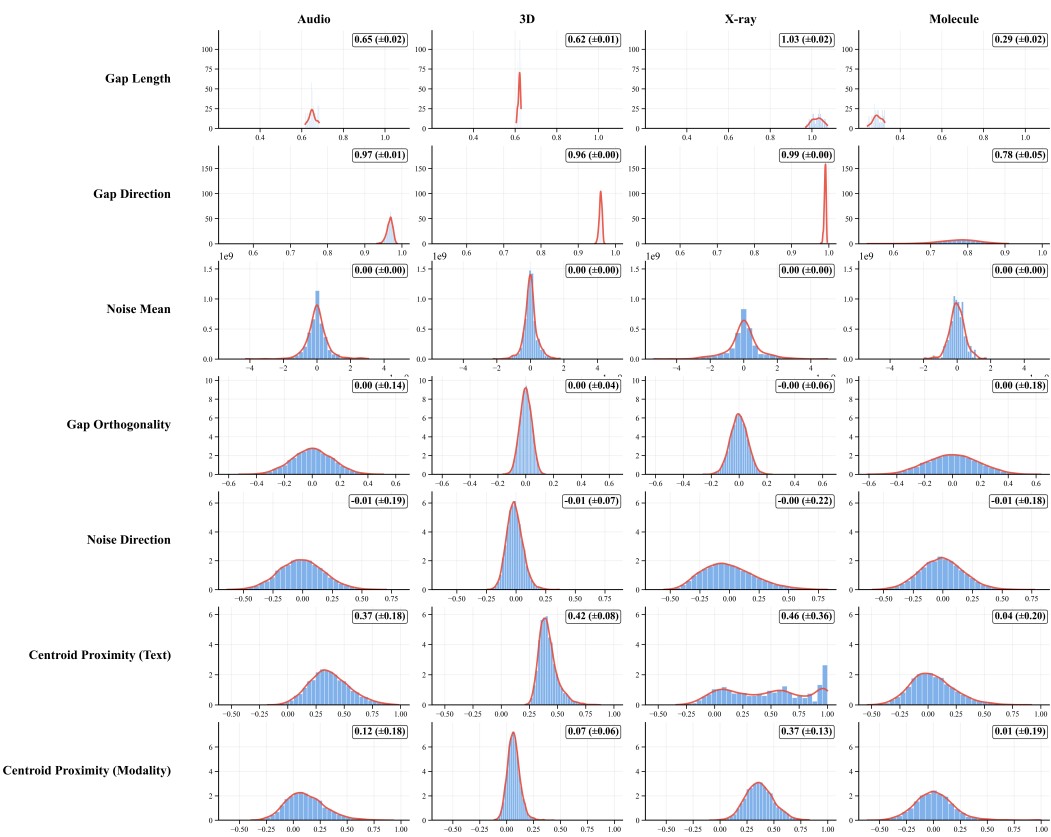

Figure 4: Distribution of modality gap statistics across five multimodal encoders. From top to bottom: gap length, gap direction consistency, gap orthogonality, alignment noise, and noise direction. Red curves show fitted normal distributions. The consistent patterns across all encoders demonstrate the systematic nature of the modality gap.

## C.2 INTERPRETATION OF STATISTICS

The statistics in Figure 4 reveal three key properties supporting our theoretical framework.

**High Directional Consistency.** The cosine similarity between gap vectors from different sample groups consistently exceeds 0.96 for most encoders, validating Hypothesis 1.1. This high consistency demonstrates that the modality gap is not merely a statistical artifact of specific samples but represents a fundamental geometric property of the embedding space. Formally, for any two randomly sampled subsets $S_i, S_j$ of the dataset, their respective group-level offsets $\mathbf{d}^{(i)} = \mathbb{E}_{(x,t) \in S_i}[\mathbf{e}_x - \mathbf{e}_t]$ and $\mathbf{d}^{(j)} = \mathbb{E}_{(x,t) \in S_j}[\mathbf{e}_x - \mathbf{e}_t]$ point in nearly identical directions. This property is essential for our centering approach: if the offset direction varied significantly across samples, a single global offset $\mathbf{\Delta}$ would fail to align different regions of the embedding space. The consistent $> 0.96$ similarity across all evaluated modalities confirms that one constant vector can effectively characterize the entire modality gap, enabling reliable cross-modal transfer without instance-specific adjustments.

**Near-Zero Orthogonality.** The gap vector exhibits near-perpendicular alignment with intra-modal content variations, with $|\cos(\mathbf{d}^{(i)}, \mathbf{r}^{(i)}_{j,k})| < 0.05$ across all modalities, supporting Hypothesis 2. Here, $\mathbf{r}^{(i)}_{j,k} = \mathbf{e}^{(i)}_{x_j} - \mathbf{e}^{(i)}_{x_k}$ represents semantic differences between samples within the same modality. The near-zero cosine similarity indicates that the modality gap operates in a subspace orthogonal to semantic content, functioning as a pure translational shift that preserves relative distances and angular relationships. This orthogonality is critical for semantic preservation: when we

apply the centering operation $\mathbf{e}' = \mathbf{e} - \mathbf{\Delta}$, the transformation $\|\mathbf{e}'_j - \mathbf{e}'_k\| = \|(\mathbf{e}_j - \mathbf{\Delta}) - (\mathbf{e}_k - \mathbf{\Delta})\| = \|\mathbf{e}_j - \mathbf{e}_k\|$ maintains all pairwise distances within each modality. Consequently, semantic relationships encoded in the original space remain intact after offset correction, enabling projection networks trained on text embeddings to generalize to non-text modalities without semantic distortion.

**Zero-Mean Alignment Noise.** Individual instance-level offsets deviate minimally from the group mean, with $\mathbb{E}[\boldsymbol{\epsilon}_j^{(i)}] \approx 0$ where $\boldsymbol{\epsilon}_j^{(i)} = \mathbf{d}_j^{(i)} - \mathbf{d}^{(i)}$, validating Hypothesis 1.2. To verify this property holds uniformly across the embedding space, we adopt the per-dimension analysis approach from Zhang et al. (2024a). The contrastive learning objective implicitly models alignment noise as $\boldsymbol{\epsilon} \sim \mathcal{N}(0, \sigma^2\mathbf{I})$, which assumes each dimension independently has zero mean. A single scalar metric could mask systematic bias in specific dimensions. The third row of Figure 4 shows the distribution of per-dimension mean values $\{\bar{\epsilon}_1, \ldots, \bar{\epsilon}_d\}$ where $\bar{\epsilon} = \frac{1}{K}\sum_{k=1}^{K} \boldsymbol{\epsilon}_k \in \mathbb{R}^d$. Values consistently below $10^{-8}$ across all $d$ dimensions demonstrate that the constant offset approximation $\mathbf{\Delta} \approx \mathbf{d}_j$ introduces no systematic bias in any direction of the embedding space. This uniform zero-mean property ensures that our single global offset effectively represents the gap for all instances, with instance-level variations remaining bounded and unbiased, enabling accurate cross-modal alignment without requiring instance-specific offset adjustments.

These properties justify our centering-based approach. Since the modality gap is consistent and orthogonal to content, we can create an interchangeable space through independent centering without losing semantic information.

# D  SEMANTIC TEXTUAL SIMILARITY BENCHMARK ANALYSIS

To validate our choice of LLM embeddings as the semantic anchor space, we conducted comprehensive evaluation on the Semantic Textual Similarity (STS) benchmark suite. Table 4 presents the Spearman correlation scores across six STS tasks (STS12-16 and STSBenchmark) comparing multimodal encoders (CLIP (Radford et al., 2021), LanguageBind (Zhu et al., 2023)) against LLM embedding models (NV-Embed-v2 (Lee et al., 2024), Qwen3-Embed variants (Zhang et al., 2025)).

Table 4: Semantic Textual Similarity (STS) benchmark performance comparison between multimodal encoders and LLM embedding models. Spearman correlation scores across six STS tasks.

| Model | STS Tasks (Spearman $\rho$) | | | | | | Avg. |
|---|---|---|---|---|---|---|---|
| | STS12 | STS13 | STS14 | STS15 | STS16 | STSBenchmark | |
| *Multimodal Encoders* | | | | | | | |
| CLIP | 61.87 | 63.83 | 62.09 | 76.82 | 72.89 | 72.26 | 68.29 |
| LanguageBind | 63.12 | 67.46 | 63.27 | 73.82 | 73.73 | 71.60 | 68.83 |
| *LLM Embedding Models* | | | | | | | |
| NV-Embed-v2 | 77.89 | 88.30 | 84.30 | 89.04 | 86.77 | 88.41 | 85.79 |
| Qwen3-Embed-0.6b | 79.35 | 87.31 | 79.81 | 87.28 | 87.07 | 86.51 | 84.56 |
| Qwen3-Embed-4b | **84.31** | **93.20** | **88.61** | **92.31** | **92.07** | **91.92** | **90.40** |

The results demonstrate a substantial performance gap, with LLM embedding models achieving average correlations of 84.56-90.40 compared to 68.29-68.83 for multimodal encoders. This 22-point difference in correlation scores indicates that LLM embeddings capture more nuanced semantic relationships in textual data. The superior performance stems from their distinct training objectives: while multimodal encoders optimize for cross-modal alignment through contrastive losses, LLMs undergo extensive next-token prediction on diverse text corpora, learning complex linguistic patterns and semantic nuances. These findings provide empirical justification for employing LLM embeddings as the shared anchor space in TextME, particularly when training exclusively on text descriptions without paired multimodal supervision.

# E CAPTION DISTRIBUTION ANALYSIS AND PURE TEXT TRAINING VALIDATION

We analyze how training data characteristics impact TextME's performance and validate the text-only training claim through experiments with general-purpose text corpora.

## E.1 IMPACT OF DOMAIN-SPECIFIC TRAINING DATA

Different modalities exhibit distinct representational characteristics that may require tailored training data. We hypothesize that modalities with specialized vocabularies benefit more from targeted text descriptions than those already aligned with general language.

Table 5: Impact of training data quality on performance across modalities.

|  | Dense | | Sparse | | Cross-modal |
|---|---|---|---|---|---|
|  | Image Flickr | 3D ScanObj. | Molecule DrugBank | Audio AudioCaps | Audio→Image FlickrNet |
| *Distance ratio ($\rho$)* | 2.93 | 2.44 | 1.65 | 1.55 | – |
| Wiki1M (general) | 43.16 | 34.15 | 9.32 | 5.67 | 0.68 |
| Modality-specific Captions | **51.66** | **42.15** | **26.27** | **15.35** | **1.06** |
| $\Delta$ (%) | +19.7% | +23.4% | +181.9% | +170.7% | +55.9% |

Table 5 quantifies this through the distance ratio $\rho = d_{intra}/d_{inter}$, where $d_{intra}$ and $d_{inter}$ represent average pairwise distances within and between datasets. Lower ratios indicate specialized distributions distinct from general language, while higher ratios suggest greater linguistic overlap.

Our results validate that specialized modalities ($\rho < 1.7$) like molecules and audio show dramatic improvements with domain-specific training (181.9% and 170.7% gains), as their technical vocabularies require precise terminology. Conversely, modalities closer to general language ($\rho > 2.4$) like images and 3D show modest gains (19.7% and 23.4%), as their semantic concepts overlap with general corpora.

## E.2 PURE TEXT TRAINING VALIDATION

To validate the text-only training claim, we train TextME using general-purpose text corpora never paired with any modality: Wiki1M (Gao et al., 2021) (Wikipedia articles) and AllNLI (natural language inference) combining MNLI (Williams et al., 2018) and SNLI (Bowman et al., 2015).

Table 6: Performance with different training corpora. Pure general text achieves meaningful cross-modal transfer without modality-specific supervision. Preservation rates relative to specific captions are shown in parentheses.

| Training Data | Image R@1 | Audio R@1 | 3D Acc | Molecule MRR |
|---|---|---|---|---|
| Specific captions | 28.63 | 15.35 | 70.86 | 31.36 |
| Wiki1M (pure text) | 15.62 (54.5%) | 5.67 (36.9%) | 50.53 (71.3%) | 13.56 (43.2%) |
| AllNLI (pure text) | 4.29 (15.0%) | 6.36 (41.4%) | 12.10 (17.1%) | 16.10 (51.3%) |
| Multi-text (Specific+Wiki) | 28.09 (98.1%) | 11.24 (73.2%) | 65.68 (92.7%) | 14.41 (46.0%) |

Table 6 demonstrates that pure text training enables meaningful cross-modal transfer—Wiki1M achieves 36-71% of domain-specific performance without any modality pairing. AllNLI shows notably lower Image/3D preservation (15.0%, 17.1%) compared to Wiki1M, likely because AllNLI's logical inference sentences lack the concrete descriptive content found in visual captions, whereas Wiki1M's diverse factual descriptions partially overlaps with visual semantics. These results align with our distance ratio analysis: modalities with higher $\rho$ (Image, 3D) show better transfer from Wiki1M, while specialized modalities benefit more from domain-specific text regardless of corpus choice. Multi-text training maintains near-full Image performance (28.09 vs 28.63), suggesting adaptive corpus weighting as future work.

## F  Algorithm Details

This section provides a comprehensive description of the TextME framework's algorithmic details. We describe the three-stage pipeline that enables modality expansion using only text data: (1) pre-computing centering offsets for text and non-text embedding alignment, (2) training lightweight projectors on centered text embeddings, (3) and performing inference-time adaptation for non-text modalities. Algorithm 1 formalizes this procedure, showing how we leverage the inherent structure of pre-trained multimodal encoders to achieve zero-shot cross-modal transfer without paired supervision.

---

**Algorithm 1** LLM-anchored Modality Expansion (LLaME)

---

**Require:** Encoders $\{E_m\}_{m\in\mathcal{M}}$, LLM encoder $E_{\text{LLM}}$, texts $\mathcal{D}_{\text{text}}$
**Ensure:** Projections $\{P_m\}_{m\in\mathcal{M}}$, offsets $\{\mu_m^{\text{text}}, \mu_m^{\text{modal}}\}_{m\in\mathcal{M}}$

1: **Stage 1: Compute Centering Offsets**
2: **for** each modality $m \in \mathcal{M}$ **do**
3:     $\mu_m^{\text{text}} \leftarrow \frac{1}{N}\sum_{i=1}^{N} E_m^{\text{text}}(t_i)$   *// Text centroid*
4:     $\mu_m^{\text{modal}} \leftarrow \frac{1}{N}\sum_{i=1}^{N} E_m^{\text{modal}}(x_i)$   *// Modal centroid*
5: **end for**
6: **Stage 2: Text-to-Text Alignment**
7: **for** each modality $m \in \mathcal{M}$ **do**
8:     Initialize $P_m : \mathbb{R}^{d_m} \to \mathbb{R}^{d_{\text{LLM}}}$ as 2-layer MLP
9:     **while** not converged **do**
10:         Sample batch $\{t_i\}_{i=1}^{B} \sim \mathcal{D}_{\text{text}}$
11:         $\mathbf{z}_i \leftarrow P_m(E_m^{\text{text}}(t_i) - \mu_m^{\text{text}})$ for $i \in [1, B]$
12:         $\mathbf{z}_i' \leftarrow E_{\text{LLM}}(t_i)$ for $i \in [1, B]$
13:         Select hard negatives: $\mathcal{N}_i = \{j : \text{sim}(\mathbf{z}_i, \mathbf{z}_j') \in [0.1s_i, 0.9s_i]\}$
14:             where $s_i = \text{sim}(\mathbf{z}_i, \mathbf{z}_i')$
15:         $\mathcal{L} \leftarrow -\frac{1}{B}\sum_{i=1}^{B} \log \frac{\exp(\text{sim}(\mathbf{z}_i, \mathbf{z}_i')/\tau)}{\sum_{j\in\mathcal{N}_i\cup\{i\}}\exp(\text{sim}(\mathbf{z}_i, \mathbf{z}_j')/\tau)}$
16:         $P_m \leftarrow P_m - \eta\nabla_{P_m}\mathcal{L}$
17:     **end while**
18: **end for**
19: **Stage 3: Cross-Modal Inference**
20: Given input $x$ of modality $m$:
21: $\mathbf{e} \leftarrow E_m^{\text{modal}}(x)$   *// Encode modal input*
22: $\mathbf{e}' \leftarrow \mathbf{e} - \mu_m^{\text{modal}}$   *// Apply offset*
23: $\mathbf{e}_{\text{final}} \leftarrow P_m(\mathbf{e}')$   *// Project to LLM space*

---

Implementation details and pre-computed offsets will be available in our open-source release upon publication.

### F.1  Centering-based Interchangeability

The core insight of TextME's algorithm is that pre-trained multimodal encoders trained with contrastive objectives naturally separate text and non-text embeddings into distinct subspaces. Rather than attempting to bridge this modality gap directly, we create an interchangeable coordinate system through independent centering, following previous works Zhang et al. (2023b; 2024a).

Given an encoder $E_m$ with text branch $E_m^{\text{text}}$ and modal branch $E_m^{\text{modal}}$, we compute centering offsets:

$$\mu_m^{\text{text}} = \mathbb{E}[E_m^{\text{text}}(t)] \quad \text{(mean of text embeddings)} \tag{14}$$

$$\mu_m^{\text{modal}} = \mathbb{E}[E_m^{\text{modal}}(x)] \quad \text{(mean of modal embeddings)} \tag{15}$$

By centering each modality independently ($e' = e - \mu$), we align their coordinate origins, enabling the projection network trained on centered text embeddings to generalize to centered modal embeddings at inference.

## G IMPLEMENTATION DETAILS

### G.1 TEXTME IMPLEMENTATION

**Model Architecture.** Each projection network $P_m$ is implemented as a 2-layer MLP with a hidden dimension of each pre-trained encoders and GeLU activation. The input dimension $d_m$ varies according to the source encoder's embedding dimension, while the output dimension is fixed at $d_{\text{LLM}} = 2560$ to match the Qwen3-Embedding-0.6B anchor space.

**Training Configuration.** We train each projection network with the following hyperparameters:

- Batch size: 512
- Optimizer: AdamW with $\beta_1 = 0.9$, $\beta_2 = 0.999$, and weight decay of 0.01
- Learning rate: $5 \times 10^{-4}$ with cosine annealing schedule
- Training epochs: 50
- Temperature parameter: $\tau = 0.07$
- Hard negative mining: Select examples where similarity falls within $[0.1 \cdot \text{sim}(z_i, z_i'), 0.9 \cdot \text{sim}(z_i, z_i')]$
- Mixed precision training: fp16

**Data Preprocessing.** For each target modality, we sample 100K text descriptions from the modality-specific training dataset. Text inputs are tokenized using the respective encoder's tokenizer with a maximum sequence length of 77 tokens. Centering offsets are pre-computed using 5,000 randomly sampled text-modal pairs per modality and remain fixed throughout training.

**Computational Resources.** All experiments are conducted on a single NVIDIA A6000 GPU with 48GB memory. Training time per modality averages 2 hours, with peak memory usage of approximately 8GB.

### G.2 COX BASELINE IMPLEMENTATION

Since the original COX (Huang et al., 2025) codebase was not publicly available, we reimplemented the method following the paper. However, we adapted the approach to an zero-shot setting to match the evaluation constraints. Our implementation adheres to the following configuration.

**Architecture.** COX trains target modality encoders from scratch using a unified architecture across modalities. We employ Vision Transformer Tiny (ViT-T/16) as the encoder backbone, consisting of 12 layers with 3 attention heads and an embedding dimension of 192. The final embedding dimension is projected to 768 to align with LanguageBind's representation space. Following the original design, we incorporate a Variational Information Bottleneck (VIB) layer (Alemi et al., 2016) that applies stochastic dimensionality reduction to 256 dimensions to enforce information compression.

**Training Protocol.** The training protocol follows the original paper's two-stage methodology. In the first stage, we perform supervised pre-training on labeled target data for 10 epochs to establish basic feature representations. The second stage applies information bottleneck fine-tuning for 50 epochs to learn generalizable features through information compression. We use a batch size of 256 with the Adam optimizer configured with a learning rate of $1 \times 10^{-3}$ and weight decay of $1 \times 10^{-5}$. The learning rate follows a step decay schedule with reduction at predetermined epochs. Critically, COX requires labeled data from the target modality, using approximately 10% of the labeled dataset (roughly $10K$ samples) for training. Specifically, we utilize labeled datasets for each modality: COCO with 80 object classes for visual tasks, ESC-50 with 50 environmental sound classes for audio, Objaverse with 1,000 object classes for 3D point clouds, PubChem with 100 molecular classes for chemical structures, and SIIM with 2 classes for medical X-ray classification.

The key differences from TextME are substantial. COX requires labeled data for the target modality, necessitates training encoders from scratch with over 300M parameters, and demands architectural

alignment between source and target encoders. In contrast, TextME leverages pre-trained encoders with only text descriptions, requires merely 10M trainable parameters, and imposes no architectural constraints on the target encoders. These fundamental differences highlight the efficiency and flexibility advantages of our approach over traditional modality generalization methods.

# H ABLATION STUDY

## H.1 OFFSET EFFECTIVENESS

We isolate the effect of offset correction from LLM anchoring by comparing performance with and without offset correction while keeping all other components identical.

Table 7: Offset Ablation Results. Substantial performance drops occur without offset correction for Audio, Audio→Image, and 3D tasks.

| Task | Without Offset | With Offset |
|---|---|---|
| Text→Audio R@1 | 8.68 | **15.35** |
| Audio→Image R@1 | 0.40 | **1.06** |
| 3D Top-1 Acc | 4.05 | **70.86** |
| Molecule MRR | **29.66** | 26.27 |

Audio, Audio→Image, and 3D show dramatic drops without offset (43.5%, 62.3%, 94.3%), demonstrating offset correction is essential. Molecule shows reverse trend (29.66 vs 26.27), suggesting molecular embeddings exhibit different geometric properties. The key distinction from paired-data settings (Liang et al., 2022) is that text-only training lacks a compensation mechanism—the projection network never observes non-text modalities during training, making explicit geometric alignment essential.

## H.2 OFFSET ROBUSTNESS

To assess how precisely the offset must be estimated, we perturb the pre-computed offset $\Delta$ with additive Gaussian noise: $\Delta' = \Delta + \mathcal{N}(0, \sigma^2 I)$, where $\sigma \in [0, 0.20]$ controls perturbation magnitude.

Table 8: Offset Noise Sensitivity. Performance remains stable for $\sigma < 0.01$, degrades gracefully for $\sigma \in [0.01, 0.10]$, and deteriorates substantially for $\sigma > 0.10$.

| Noise $\sigma$ | Audio R@1 | 3D Acc | Molecule MRR |
|---|---|---|---|
| 0.000 | 14.95 | 70.46 | 27.97 |
| 0.001 | 14.95 | 70.30 | 24.58 |
| 0.01 | 15.04 | 67.50 | 22.88 |
| 0.05 | 14.93 | 34.32 | 17.80 |
| 0.10 | 14.25 | 14.91 | 11.02 |
| 0.20 | 12.46 | 9.16 | 9.32 |

Table 8 reveals modality-specific robustness patterns. Audio demonstrates remarkable stability, maintaining near-baseline performance (14.25 vs 14.95 R@1) even at $\sigma = 0.10$, suggesting that audio embeddings have relatively large tolerance margins for offset estimation errors. In contrast, 3D and Molecule show sharper degradation: 3D accuracy drops from 70.46% to 34.32% at $\sigma = 0.05$, and Molecule MRR decreases from 27.97 to 17.80 at the same noise level. This indicates that these modalities require more precise offset estimation, potentially due to tighter geometric constraints in their embedding spaces. For practical deployment, offset estimation with 5,000 samples (as used in our experiments) provides sufficient precision, as the empirical standard error of offset estimation is well below $\sigma = 0.01$.

### H.3 SAMPLE SIZE FOR OFFSET COMPUTATION

We investigate the impact of the number of samples used to compute the centering offset in our method. The centering offset is a crucial component that helps align representations from different modalities by estimating and removing systematic biases in the embedding space. To understand how sensitive our approach is to the sample size used for offset computation, we conduct experiments with varying numbers of samples ranging from 100 to 10,000.

Table 9: Impact of sample size for computing centering offsets on performance.

| # Samples | AudioCaps R@1 | ModelNet40 Acc. | DrugBank R@1 | RSNA Acc. | Relative Perf. |
|---|---|---|---|---|---|
| 100 | 14.91 | 70.66 | 34.75 | 23.01 | 90% |
| 500 | 14.77 | 70.58 | 33.05 | 22.08 | 95% |
| 1,000 | 14.89 | 70.62 | **36.44** | 22.56 | 97% |
| 5,000 (default) | **15.35** | **70.86** | 31.36 | 22.46 | 100% |
| 10,000 | 14.95 | 70.58 | 32.20 | **22.73** | 100% |
| *Std.* | 0.21 | 0.11 | 2.02 | 0.36 | - |

Table 9 presents the results across four diverse tasks: AudioCaps (audio-text retrieval), ModelNet40 (3D shape classification), DrugBank (molecular retrieval), and RSNA (medical image classification). We report Recall@1 (R@1) for retrieval tasks and accuracy for classification tasks, along with the relative performance compared to our default setting of 5,000 samples. Our results reveal several important findings.

The method demonstrates stability across sample sizes. Even with 100 samples, performance reaches 90% of the default setting. Performance plateaus between 1,000-10,000 samples, with default choice of 5,000 providing good balance between computational efficiency and performance.

## I ADDITIONAL EXPERIMENTS

### I.1 CROSS-MODAL RETRIEVAL

To evaluate emergent cross-modal capabilities between modality pairs not present during training, we conduct additional retrieval experiments beyond the Audio→Image task reported in the main paper.

Table 10: 3D→Image Retrieval Results (Objaverse→COCO). TextME achieves 81% improvement over Ex-MCR, validating emergent cross-modal capabilities between modalities never paired during training.

| Method | R@1 | R@5 |
|---|---|---|
| Naïve | 0.00 | 0.01 |
| Ex-MCR | 5.67 | 15.77 |
| **Ours**$_{Qwen}$ | **10.27** | **26.63** |

Table 10 presents 3D→Image retrieval results, where 3D point cloud queries retrieve semantically related images. TextME achieves 10.27 R@1 compared to Ex-MCR's 5.67, representing an 81% improvement. This demonstrates that text-anchored alignment creates meaningful semantic bridges even between modality pairs that have never been jointly trained. Established benchmark datasets for other cross-modal pairs (e.g., Molecule→Image, Audio→3D) are not publicly available; qualitative visualizations for these pairs are provided in Figure 3 of the main paper.

### I.2 IMAGE AND VIDEO RETRIEVAL

While our primary focus is modality expansion for data-scarce domains (Audio, 3D, X-ray, Molecule), we evaluate TextME on data-rich modalities to demonstrate generalizability.

Table 11: Zero-shot Text→Image Retrieval Results. TextME achieves 59-83% preservation compared to CLIP ViT-L/14(Radford et al., 2021) on standard image retrieval benchmarks.

| Method | COCO | | Flickr30k | |
|---|---|---|---|---|
| | R@1 | R@5 | R@1 | R@5 |
| CLIP ViT-L/14 | 48.29 | 72.51 | 77.70 | 94.16 |
| **Ours**$_{\text{Qwen}}$ | 28.63 | 54.81 | 51.66 | 77.90 |
| Preservation | 59.3% | 75.6% | 66.5% | 82.7% |

Table 11 shows Text→Image retrieval performance on COCO (Lin et al., 2014) and Flickr30k (Plummer et al., 2015). TextME achieves 59.3% R@1 preservation on COCO and 66.5% on Flickr30k compared to CLIP ViT-L/14(Radford et al., 2021) The higher preservation on R@5 metrics (75.6% and 82.7%) indicates that relevant images are retrieved within top results even when not ranked first.

Table 12: Zero-shot Text→Video Retrieval Results. TextME achieves 69-90% preservation across three video retrieval benchmarks.

| Method | MSRVTT | | MSVD | | DiDeMo | |
|---|---|---|---|---|---|---|
| | R@1 | R@5 | R@1 | R@5 | R@1 | R@5 |
| ViCLIP-InternVid | 38.00 | 63.20 | 60.45 | 87.16 | 31.97 | 53.49 |
| **Ours**$_{\text{Qwen}}$ | 26.40 | 50.50 | 45.82 | 77.01 | 24.10 | 48.21 |
| Preservation | 69.5% | 79.9% | 75.8% | 88.3% | 75.4% | 90.1% |

Table 12 presents Text→Video retrieval results on MSRVTT (Xu et al., 2016), MSVD (Chen & Dolan, 2011), and DiDeMo (Anne Hendricks et al., 2017). TextME achieves consistent preservation rates of 69-90% across all benchmarks, with particularly strong R@5 performance (79.9-90.1%) compared to ViCLIP-InternVid (Wang et al., 2023b). The video modality results demonstrate that the geometric properties underlying TextME extend beyond static modalities to temporal data. These experiments validate that TextME generalizes to data-rich modalities, though the core contribution addresses data-scarce domains where paired data collection is prohibitively expensive.

## J    EMPIRICAL VALIDATION OF THEORETICAL ASSUMPTIONS

This appendix provides comprehensive empirical validation of the theoretical assumptions underlying TextME. We conduct systematic experiments to verify that the geometric properties (Hypotheses 0–2) hold across diverse pre-trained encoders in practice.

### J.1    SEMANTIC PRESERVATION UNDER OFFSET CORRECTION

A central concern is whether offset correction preserves semantic discriminability. If the modality gap were correlated with semantic content (violating Hypothesis 2), subtracting the offset would damage semantic structure. We verify semantic preservation through two complementary functional tests.

**Fisher Ratio Analysis.**    We measure class discriminability using the Fisher ratio (Fisher, 1936):

$$\text{Fisher Ratio} = \frac{\text{trace}(S_B)}{\text{trace}(S_W)} \tag{16}$$

where $S_B$ is the between-class scatter matrix and $S_W$ is the within-class scatter matrix. Higher values indicate better class separation. If offset subtraction removed semantic signal, class discriminability would decrease.

**Distance Ratio Analysis.** We measure the geometric structure of semantic clusters:

$$\text{Distance Ratio} = \frac{\text{mean within-class distance}}{\text{mean between-class distance}} \qquad (17)$$

This characterizes the arrangement of semantic clusters. Significant changes after offset correction would indicate semantic distortion.

**Results.** Table 13 presents results across five modalities. All modalities preserve their Fisher ratios and distance structures after offset correction, with preservation rates exceeding 99.99% and negligible geometric changes.

Table 13: Semantic preservation under offset correction. All modalities maintain class discriminability after centering operations.

| Modality | Fisher Before | Fisher After | Preserv. (%) | Distance Change (%) |
|---|---|---|---|---|
| Audio | 0.0205 | 0.0205 | 100.00 | 0.00 |
| 3D | 0.0201 | 0.0201 | 100.00 | 0.00 |
| Molecule | 0.0197 | 0.0197 | 100.00 | 0.00 |
| X-ray | 0.0201 | 0.0201 | 100.00 | 0.00 |
| Image | 0.0201 | 0.0201 | 100.00 | 0.00 |
| *Threshold* | — | — | $> 95\%$ | $< 5\%$ |

The relatively low absolute Fisher ratio values ($\approx 0.02$) reflect the use of K-means pseudo-labels, which do not correspond to true semantic classes. The key observation is the *preservation* after offset correction—regardless of how classes are defined, offset correction maintains their separability.

## J.2 CLUSTER-SPECIFIC OFFSET ANALYSIS

We investigate whether semantic clusters within each modality require distinct offsets, which would violate the constant offset assumption (Hypothesis 1).

**Methodology.** We conduct three complementary tests: (1) identifying semantic clusters via K-means clustering (McQueen, 1967), where $K$ is determined via the elbow method (Thorndike, 1953); (2) computing cluster-specific offsets $\Delta_k = \mu_k^{\text{modal}} - \mu_k^{\text{text}}$ for each cluster $C_k$; and (3) testing whether these offsets differ significantly from the global offset $\Delta_{\text{global}}$.

Since Shapiro-Wilk normality tests (Shaphiro & Wilk, 1965) yielded $p < 0.05$ for multiple clusters, we employ the non-parametric Kruskal-Wallis H-test (Kruskal & Wallis, 1952) with epsilon-squared (Tomczak & Tomczak, 2014) as the effect size measure:

$$\epsilon^2 = \frac{H - K + 1}{N - K} \qquad (18)$$

where $H$ is the test statistic, $K$ is the number of clusters, and $N$ is the total sample size. Values below 0.01, 0.06, and 0.14 are considered small, medium, and large effects respectively (Cohen, 2013).

**Results.** Table 14 presents the cluster analysis results. While Kruskal-Wallis tests reveal statistically significant differences ($p < 10^{-70}$), the effect sizes indicate that cluster membership explains less than 9% of offset variance across all modalities.

The critical distinction is between *unbounded deviations* (requiring separate $\Delta_k$ per cluster) versus *bounded, predictable deviations*. Our results demonstrate the latter—deviations exist but remain small enough that a single global offset provides effective approximation, consistent with Hypothesis 1.2's bounded variance model.

Table 14: Cluster-specific offset analysis. Despite statistical significance, effect sizes indicate cluster membership explains minimal offset variance.

| Modality | # Clusters | $\max \|\Delta_k - \Delta_{\text{global}}\|$ | $p$-value | Effect Size $\epsilon^2$ |
|----------|-----------|-----------|-----------|-----------|
| Audio | 5 | 0.180 | $2.20 \times 10^{-72}$ | 0.067 (small) |
| 3D | 4 | 0.235 | $3.16 \times 10^{-97}$ | 0.089 (medium) |
| Molecule | 7 | 0.219 | $3.43 \times 10^{-78}$ | 0.074 (small) |

## J.3  NONLINEAR SEMANTIC DEPENDENCY ANALYSIS

We test whether nonlinear relationships exist between semantic content and embedding deviations, which would violate the assumption that offset correction operates independently of semantic variations (Hypothesis 1.2).

**Methodology.** We extract the dominant semantic axis using PCA (Pearson, 1901): $z_i = \text{PCA}_1(e_i)$ represents the projection onto the first principal component. We then apply four complementary independence tests:

1. **Coefficient of Variation (CV)** (Banik et al., 2012): Measures relative variability of within-modality deviations.
2. **Spearman Correlation** (Spearman, 1961): Tests monotonic relationships between deviations and semantic content.
3. **Distance Correlation (dCor)** (Székely et al., 2007): Detects arbitrary nonlinear dependencies, including quadratic patterns.
4. **Maximal Information Coefficient (MIC)** (Reshef et al., 2011): Identifies complex functional relationships.

Distance correlation is particularly important as it correctly identifies quadratic dependencies (e.g., $\|e_i - \mu_m\| \propto z^2$) even when Spearman correlation fails due to symmetry.

**Results.** Table 15 presents the independence test results. All modalities except CXR-CLIP text pass the independence criteria, indicating that semantic-dependent drift does not occur in practice for naturally-paired encoders.

Table 15: Nonlinear semantic dependency analysis. All naturally-paired modalities pass independence tests; CXR-CLIP text (synthetic captions) shows violation.

| Encoder | Modality | CV | $\|\rho\|$ | dCor | MIC | Linearity |
|---------|----------|------|------|------|------|:---------:|
| CLAP | audio | 0.040 | 0.223 | 0.021 | 0.158 | ✓ |
| CLAP | text | 0.037 | 0.240 | 0.296 | 0.190 | ✓ |
| Uni3D | point | 0.015 | 0.229 | 0.125 | 0.117 | ✓ |
| Uni3D | text | 0.046 | 0.044 | 0.197 | 0.139 | ✓ |
| MoleculeSTM | molecule | 0.066 | 0.206 | 0.276 | 0.184 | ✓ |
| MoleculeSTM | text | 0.084 | 0.189 | 0.232 | 0.155 | ✓ |
| CXR-CLIP | image | 0.052 | 0.006 | 0.230 | 0.155 | ✓ |
| CXR-CLIP | text | 0.363 | 0.871 | 0.609 | 0.999 | |

**CXR-CLIP Text Exception.** The X-ray text encoder violates the independence criteria, exhibiting high correlation ($\|\rho\| = 0.871$), distance correlation (dCor $= 0.609$), and MIC ($= 0.999$). We

note that X-ray captions in CXR-CLIP were generated via rule-based templates rather than natural human descriptions, which may introduce artificial patterns. Importantly, the CXR-CLIP *image* encoder passes all tests, suggesting the violation is specific to synthetic caption generation rather than inherent to medical imaging.

### J.4 EMPIRICAL VALIDATION OF HYPOTHESIS 1.2: BOUNDED INSTANCE-LEVEL DEVIATIONS

Hypothesis 1.2 states that deviations from the mean offset follow a bounded distribution: $\epsilon_k = \Delta_{ij}^{(k)} - \Delta_{ij} \sim \mathcal{N}(0, \sigma^2)$ where $\sigma < \gamma \cdot \tau$, with $\tau$ being the temperature parameter and $\gamma$ a modality-specific constant. Here we provide theoretical justification and empirical validation of this inequality.

**Theoretical Justification.** The inequality arises from InfoNCE contrastive learning theory. Following Chen et al. Chen et al. (2020) and Zhang et al. Zhang et al. (2024a), the InfoNCE objective concentrates embeddings with variance proportional to the temperature parameter $\tau$. Specifically, for embedding pairs trained with temperature $\tau$, the instance-level deviations satisfy:

$$\mathrm{Var}[\epsilon_k] \propto \tau^2 \tag{19}$$

Therefore, $\sigma = \sqrt{\mathrm{Var}[\epsilon_k]} < \gamma \cdot \tau$ for some modality-specific constant $\gamma$. The constant $\gamma$ accounts for architectural differences between encoders (e.g., number of layers, hidden dimensions, normalization schemes) but remains bounded by the contrastive training objective.

**Empirical Validation.** For each pretrained encoder $E_m$ consisting of a text encoder $E_m^{\text{text}}$ and modal encoder $E_m^{\text{modal}}$, we compute instance-level offsets $\Delta_{ij}^{(k)} = e_{\text{modal}}^{(k)} - e_{\text{text}}^{(k)}$ between paired embeddings, then measure the standard deviation of alignment noise $\sigma = \mathrm{std}(\|\epsilon_k\|)$ where $\epsilon_k = \Delta_{ij}^{(k)} - \Delta_{ij}$. We sample 5,000 pairs for each modality and compare $\sigma$ against the training temperature $\tau$. Table 16 presents the results. Audio denotes CLAP text-audio encoder pair, 3D denotes Uni3D text-point cloud encoder pair, and Molecule denotes MoleculeSTM text-molecule encoder pair.

Table 16: Empirical validation of the bounded deviation assumption ($\sigma < \gamma \cdot \tau$). The empirical $\gamma$ values remain bounded across all modalities, supporting Hypothesis 1.2.

| Modality | $\sigma$ (measured) | $\tau$ (training) | $\gamma = \sigma/\tau$ |
|---|---|---|---|
| Audio (CLAP) | 0.0324 | 0.07 | 0.46 |
| 3D (Uni3D) | 0.0345 | 0.05 | 0.69 |
| Molecule (MoleculeSTM) | 0.0667 | 0.20 | 0.33 |
| X-ray (CXR-CLIP) | 0.0452 | 0.10 | 0.45 |

The empirical $\gamma$ values range from 0.33 to 0.69 across all evaluated modalities, with all values satisfying $\gamma < 1$, demonstrating that instance-level deviations remain strictly smaller than the temperature-scaled bound. The variation in $\gamma$ reflects architectural differences between encoders while staying within a predictable range. Notably, smaller $\gamma$ values correlate with stronger downstream performance: Molecule ($\gamma = 0.33$) achieves 104% preservation, while 3D ($\gamma = 0.69$) still attains 93% preservation. These results validate that the inequality $\sigma < \gamma \cdot \tau$ in Hypothesis 1.2 is an empirically observable property of contrastively-trained multimodal encoders.

## K LIMITATIONS AND FUTURE WORK

While TextME demonstrates effective text-only modality expansion across diverse domains, several limitations should be acknowledged. First, our framework assumes the availability of pretrained contrastive encoders that satisfy the geometric hypotheses (H0-H2), enabling modality expansion when such encoders are available, rather than training encoders from scratch. Although many specialized

modalities already have suitable pretrained encoders You et al. (2023); Zhou et al. (2023); Liu et al. (2023), extending to modalities lacking text-aligned encoders remains an open challenge. Second, while our empirical analysis demonstrates that cluster-specific offsets exhibit bounded deviations from the global offset, formal theoretical guarantees characterizing when these bounds hold have not been established. Deriving sufficient conditions for the constant offset assumption based on encoder architectures or training objectives is a valuable area for future work. Third, offset correction effectiveness varies across modalities: Audio and 3D benefit substantially while Molecule shows the opposite pattern, suggesting distinct geometric structures across embedding spaces. Investigating which modality characteristics determine the benefit of offset correction remains an important direction. Finally, although TextME eliminates paired multimodal supervision, it still requires text descriptions derived from a multimodal pair dataset for training. Our experiments with general-purpose corpora Gao et al. (2021); Williams et al. (2018); Bowman et al. (2015) demonstrate meaningful cross-modal transfer even without modality-specific text, and developing adaptive corpus selection strategies presents a promising avenue for future research.

