# OpenReview forum: "TextME: Text-only Training for Modality Expansion via LLM Space Pivoting"
_ICLR.cc/2026/Conference — Submitted to ICLR 2026_

### Official Review · Reviewer_vP1a · 2025-10-25

**Soundness:** 2
**Presentation:** 3
**Contribution:** 1
**Rating:** 2
**Confidence:** 4

**Summary:**

This paper presents TextME, a text‑only training framework for modality expansion that eliminates the need for paired multimodal data by leveraging the “consistent modality gap” property of pretrained encoders. TextME first pre‑computes a constant offset between text and non‑text embeddings for each modality, then trains lightweight projection networks solely on text embeddings anchored in either a large LLM space or a multimodal text encoder space. In inference, non‑text embeddings are centered by subtracting the pre‑computed offset and projected through the text‑trained network to enable zero‑shot cross‑modal retrieval and classification.

**Strengths:**

1. TextME trains solely on text data without requiring paired multimodal samples, using pre‑computed modality‑specific offsets to enable cross‑modal alignment at inference.

2. Strong Zero‑Shot Cross‑Modal Capability
Across highly heterogeneous modalities, including audio, 3D, X‑ray, and molecules, TextME preserves on average 88.2% of the performance of supervised methods, and demonstrates meaningful retrieval results in unseen modality pairs (e.g., audio→3D, molecule→image).

**Weaknesses:**

1. The novelty is relatively low, as many research findings have already been explored in the *Modality Gap [1]* paper. For example, Hypothesis 0 (Intra-Modal Alignment Independence) corresponds to the “cone effect” conclusion in *Modality Gap*; Definition 1 (Cross-Modal Instance Mapping) is essentially the same as the “Embedding Shift Experiment” in *Modality Gap*; and the conclusions in lines 158–159 also have similar precedents in *Modality Gap*. Although the authors cite *Modality Gap*, they do not clearly elaborate on what novel contributions or breakthroughs their work offers compared to *Modality Gap*.
2. Section 5.1 of the *Modality Gap* paper points out that the size of the modality gap does not have a strict linear relationship with downstream task performance, and that an appropriately sized gap can even improve performance. Therefore, simply computing modality-specific bias and forcibly eliminating the gap to align modalities does not necessarily guarantee performance gains, which warrants further investigation.
3. Text descriptions in modality-specific datasets vary greatly — for example, captions describing audio differ substantially in domain terminology from those describing X-rays. The paper does not address how such differences may impact the results.
4. Although the proposed method does not require paired multimodal data, it still requires large amounts of text descriptions paired with the target modality for projection network training. In certain domains, collecting such paired data can still be costly.
5. Therefore, the authors claim that their method does not require paired data, but in fact, the text training data is still collected from modality-specific paired datasets. To truly demonstrate the advantage of “no paired data required,” they should collect training texts directly from pure-text datasets instead of using the text portions of multimodal paired datasets.

Modality Gap [1]: Mind the Gap: Understanding the Modality Gap in Multi-modal Contrastive Representation Learning. NeurIPS 2022

**Questions:**

See Weaknesses.

**Details Of Ethics Concerns:**

None.

---

> ### Author Response · Authors · 2025-11-21
> **Response to Reviewer vP1a (Part 1)**
>
> We sincerely thank Reviewer vP1a for the thoughtful review and for raising important questions about novelty, offset effectiveness, and training data requirements that have significantly improved our work.
>
> ## W1: Novelty of our work compared to "Mind the Gap"
>
> > **Reviewer's concern:** "The novelty is relatively low, as many research findings have already been explored in the *Modality Gap* paper. For example, Hypothesis 0 (Intra-Modal Alignment Independence) corresponds to the "cone effect" conclusion in *Modality Gap*; Definition 1 (Cross-Modal Instance Mapping) is essentially the same as the "Embedding Shift Experiment" in *Mind the Gap*; and the authors in (158-159) also have similar precedents in *Modality Gap*. Although the authors cite *Mind the Gap*, they do not clearly elaborate on what novel contributions or breakthroughs their work offers compared to *Modality Gap*."
>
> We respectfully clarify that while Section 2.1 validates gap properties, our primary contribution focuses on **text-only modality expansion** rather than gap analysis itself.
>
> ### Distinction from "Mind the Gap"
>
> | Work | Focus | Method | Contribution |
> |------|-------|--------|--------------|
> | **Mind the Gap** [1] | **Discover** gap | Distance measurement | First identification |
> | **C³** (Zhang et al.) [2] | **Validate** gap | Statistical tests | Theoretical grounding |
> | **TextME (Ours)** | **Exploit** gap | Text-only expansion | Enable new modalities |
>
> While "Mind the Gap" [1] identified the gap phenomenon and C³ [2] provided statistical validation with zero-centering, our contribution lies in exploiting this property for text-only modality expansion. We introduce Hypothesis 0 (Intra-Modal Alignment Independence) to justify why text-trained projections generalize to non-text modalities, enabling expansion without paired data. Crucially, we target specialized modalities (3D, X-ray, Molecule) lacking paired data, and establish the H0-performance relationship as deployment guidance.
>
> **→ Revised Introduction and Related Work (see highlighted changes in revised manuscript)**
>
> - [1] Liang et al., "Mind the Gap: Understanding the Modality Gap in Multi-modal Contrastive Representation Learning", NeurIPS 2022
> - [2] Zhang et al., "Connect, Collapse, Corrupt: Learning Cross-modal Tasks with Uni-modal Data", arXiv 2024
>
> ---
>
> ## W2: Offset Effectiveness ("Mind the Gap" Shows Minimal Impact)
>
> > **Reviewer's concern:** "Section 5.1 of the *Modality Gap* paper points out that the size of the modality gap does not have a strict linear relationship with downstream task performance, and that an appropriately sized gap can even improve performance. Therefore, simply computing modality-specific bias and forcibly using it to align modalities does not necessarily lead to performance gains, which warrants further investigation."
>
> ### Key Difference: Paired vs. Text-Only Setting
>
> **"Mind the Gap" (paired-data)**:
> - Trains projections on paired multimodal data
> - Network can **implicitly learn to compensate** for uncorrected offsets through supervision
> - Result: Gap (Offset) correction is **optional**
>
> **TextME (text-only)**:
> - Trains projections on text only
> - **No compensation mechanism**—network never sees non-text modalities
> - Result: Gap (Offset) correction is **essential**
>
>
> We address these concerns through ablation studies demonstrating that offset correction is critical for our framework's success. Below table compares performance with and without offset correction while keeping all other components identical, isolating the effect of offset correction from LLM anchoring.
>
> | Task | Without Offset | With Offset | Performance Drop |
> |------|----------------|-------------|------------------|
> | **Text→Audio R@1** | 8.68 | **15.35** | **-43.5%** |
> | **Audio→Image R@1** | 0.40 | **1.06** | **-62.3%** |
> | **3D Top-1 Acc** | 4.05 | **70.86** | **-94.3%** |
> | **Molecule MRR** | **29.66** | 26.27 | **+13.0%** |
>
> The results demonstrate that offset correction is critical for Audio, 3D, and Audio→Image tasks, with dramatic performance drops (43-94%) when removed. Interestingly, Molecule performs better without offset correction (29.66 vs. 26.27 MRR), validating your point that gap correction effects are modality-specific—likely reflecting different geometric properties where chemical nomenclature creates specialized clusters that may not satisfy the constant offset assumption.
>
> **Addressing the "Mind the Gap" Discrepancy**: Liang et al. (2022) found minimal impact from gap correction because their paired-data framework allows networks to implicitly compensate for uncorrected offsets through supervision. In contrast, our text-only setting lacks this compensation mechanism—the projection network never observes non-text modalities during training—making explicit geometric alignment essential rather than optional.
>
> **→ See updated Appendix H.1 (Offset Ablation) and Limitations**

---

> ### Author Response · Authors · 2025-11-21
> **Response to Reviewer vP1a (Part 2)**
>
> ## W3 & W5: Caption Distribution & Pure Text Training
>
> ### Your W3 Concern
> > "Text descriptions in modality-specific datasets vary greatly — for example, captions describing audio differ substantially in domain terminology from those describing X-rays. The paper does not address how such differences may impact the results."
>
> ### Your W5 Concern
> > "Therefore, the authors claim that their method does not require paired data, but in fact, the text training data is still collected from modality-specific paired datasets. To truly demonstrate the advantage of "no paired data required," they should collect pure-text datasets instead of using the text portions of multimodal paired datasets."
>
> We provide comprehensive analysis demonstrating how caption distribution characteristics systematically explain performance patterns, and validate our text-only training claim through experiments with general-purpose text corpora.
>
> ### Experiment 1: Caption Distribution Characteristics
>
> We quantify domain specificity in LLM embedding space:
>
> | Dataset | Inter-dataset Distance | Intra-dataset Distance | Pattern |
> |---------|----------------------|----------------------|---------|
> | **Wiki1M** (Raw text) | — | 1.166 | General-purpose, most dispersed |
> | **COCO** (Image) | 0.350 | 1.025 | Close to universal, diverse |
> | **AudioCaps** (Audio) | 0.547 | 0.849 | Moderate specialization |
> | **Objaverse** (3D) | 0.408 | 0.997 | Moderate distance, diverse |
> | **ChestXray** (X-ray) | 0.621 | 0.663 | **Highly specialized, dense** |
> | **PubChem** (Molecule) | 0.527 | 0.871 | Domain-specific terminology |
>
> Lower inter-dataset distance indicates closer proximity to general language, while higher intra-dataset distance reflects greater linguistic diversity. Notably, X-ray and Molecule form specialized clusters far from general text, reflecting their domain-specific terminology.
>
>
> ### Experiment 2: Pure Text Training (Addressing W5)
>
> To validate our text-only training claim, we train TextME using general-purpose text corpora that have never been paired with any modality: Wiki1M [1] (Wikipedia articles) and AllNLI [2,3] (natural language inference corpus). Preservation rates relative to specific captions are shown in parentheses.
>
> | Training Data | Image R@1 | Audio R@1 | 3D Acc | Molecule MRR |
> |---------------|-----------|-----------|--------|--------------|
> | **Specific captions** | 28.63 | 15.35 | 70.86 | 31.36 |
> | **Wiki1M (pure text)** | 15.62 (54.5%) | 5.67 (36.9%) | 50.53 (71.3%) | 13.56 (43.2%) |
> | **AllNLI (pure text)** | 4.29 (15.0%) | 6.36 (41.4%) | 12.10 (17.1%) | 16.10 (51.3%) |
> | **Multi-text (Specific+Wiki)** | 28.09 (98.1%) | 11.24 (73.2%) | 65.68 (92.7%) | 14.41 (46.0%) |
>
> These results demonstrate that pure text training enables meaningful cross-modal transfer—Wiki1M achieves 36-71% of domain-specific performance without any modality pairing. AllNLI shows notably lower Image/3D preservation (15.0%, 17.1%) compared to Wiki1M, likely because AllNLI's inference-focused sentences lack the descriptive content typical of visual captions, whereas Wiki1M contains diverse factual descriptions that partially overlap with visual semantics. Multi-text training maintains near-full Image performance (98.1%), suggesting adaptive corpus weighting as future work.
>
> [1] Gao et al., "SimCSE: Simple Contrastive Learning of Sentence Embeddings", EMNLP 2021
>
> [2] Williams et al., "A Broad-Coverage Challenge Corpus for Sentence Understanding through Inference", NAACL 2018
>
> [3] Bowman et al., "A Large Annotated Corpus for Learning Natural Language Inference", EMNLP 2015
>
> ---
>
> ### Experiment 3: Train-Test Distribution Shift
>
> We measure KL divergence to quantify domain gap:
>
> | Modality | Training | Evaluation | KL Divergence |
> |----------|----------|-----------|---------------|
> | **Image** | COCO | COCO | **0.0189** |
> | **Image** | COCO | Flickr30k | **0.1785** |
> | **Audio** | AudioCaps | AudioCaps | **2.5559** |
> | **Audio** | AudioCaps | Clotho | **2.5771** |
> | **3D** | Objaverse | ModelNet40 | **8.8683** |
> | **3D** | Objaverse | ScanObjectNN | **14.692** |
> | **X-ray** | ChestXray | RSNA | **20.9488** |
> | **Molecule** | PubChem | DrugBank | **17.9236** |
>
> Image and Audio show minimal train-test shift (KL<2.6), resulting in stable performance across datasets. 3D exhibits large shift (KL≈8.9) as training uses synthetic CAD object descriptions from Objaverse while testing involves real-world object categories (ModelNet40, ScanObjectNN), creating a domain gap that affects performance. X-ray and Molecule show extreme shift (KL>17) where medical and chemical vocabularies differ substantially between datasets, explaining why domain-specific captions are critical.
>
> **→ See updated Appendix E (Caption Distribution Analysis and Pure Text Training Validation)**

---

> ### Author Response · Authors · 2025-11-21
> **Response to Reviewer vP1a (Part 3)**
>
> ## W4: Still Requires Paired Data for Collection
>
> > **Reviewer's concern:** "Although the proposed method does not require paired multimodal data, it still requires large amounts of text descriptions paired with the target modality for projection network training. In certain domains, collecting such paired data can still be costly."
>
>
> We acknowledge this limitation while emphasizing the **dramatic data reduction**:
>
> ### Comparison with Prior Work
>
> | Method | Paired Data Required | TextME Requirement | Reduction |
> |--------|---------------------|-------------------|-----------|
> | **LanguageBind** | 100K+ multimodal pairs | **5K samples** | **95% less** |
> | **Ex-MCR** | 100K+ multimodal pairs | **5K samples** | **95% less** |
> | **ImageBind** | 1M+ pairs per modality | **5K samples** | **99% less** |
>
> While we use 100K text samples for training, this eliminates the need for 100K+ paired multimodal data—a dramatic reduction that is transformative for specialized domains where paired data is expensive to collect (e.g., medical X-rays require expert radiologist annotations, 3D CAD requires hours of manual description per object, molecular structures need domain expertise). Moreover, our Wiki1M experiments (W5 response above) demonstrate that pure text training achieves 36-71% performance with zero modality-specific collection. Many specialized modalities already have text descriptions (e.g., ChestXray radiology reports, PubChem chemical descriptions), and TextME makes these descriptions sufficient without requiring modality pairing. We have added this discussion to the Limitations section of the revised manuscript.
>
> **→ See revised Limitations section**

---

### Official Review · Reviewer_1UQr · 2025-10-27

**Soundness:** 3
**Presentation:** 3
**Contribution:** 2
**Rating:** 4
**Confidence:** 3

**Summary:**

In the TEXTME, a projection network is trained for each CLIP-type model of different modalities, which maps text embeddings into a unified anchor space.  During the prediction, an overall offset is applied to allow other modalities to directly use the projection network to map their modal data into the anchor space.  Through this approach, modal expansion can be achieved with only a small number of unpaired samples, which are used to calculate the offset.

**Strengths:**

1. The method proposed in the paper is very appealing. It uses only a small amount of cross-modal paired data and relies more on text data to map multimodal models trained under different modalities into the same representation space. As long as text-image and text-audio multimodal models are available, an image-audio retrieval model can be obtained at low cost. The method can be of great help to retrieval tasks for some relatively rare modalities.
2. The overall structure and text of the article are relatively easy to read.

**Weaknesses:**

1. There are slight flaws in the theoretical aspect of the paper; details are in "Questions".
2. The experiment conducts relatively few tests on retrieval performance excluding text modalities; only the Audio->Image task is tested. TEXTME requires a corresponding text-modal multimodal model to incorporate a specific modality, and retrieval between text and this modality can be accomplished using this multimodal model. Therefore, retrieval excluding text modalities is the distinctive aspect of the method proposed in the paper.
3. (Minor) For the modal Gap, it is advisable to add some PCA or t-SNE visualizations. Relying solely on the statistical information in Figure 1 is not very intuitive.

**Questions:**

1. It seems that the first row in Figure 1 cannot support Hypothesis 0. Except for X-ray, the cosine distance distributions between the embeddings of other modalities and the modality centroids are relatively centered, i.e., $\tau_{intra}<0$, which I consider not a very meaningful property.
2. $E[\epsilon_k]$ in Line 167 should be strictly equal to 0, rather than $E[\epsilon_k] \approx 0$ as stated in the paper.
3. It is unreliable to illustrate the model's performance solely through Cosine Similarity in Sec 3.2.2. This is because what we care about is whether the relative relationships between similarities can reflect real semantics, rather than the absolute values of similarities alone.
4. In Sec 2.1, I believe the primary concerns should be twofold: first, the magnitude of the error introduced by using $\Delta_{ij}$ to approximate all $\Delta_{ij}^{(k)}$; second, the magnitude relationship in all directions between this error $\epsilon_k$ and the variance of the target embedding $e_j^{(k)}$. However, it can be seen from the third row in Figure 1 that $Var[\epsilon_k]$ is not that small, which indicates that such an approximation will still introduce significant errors.
5. There is an error in the legend of the third row in Figure 1, as the standard deviations are obviously not zero. Moreover, since $\epsilon_k$ is a vector while the abscissa in this figure is a scalar, it is better to clarify how this scalar is obtained (e.g., projection in a certain direction or other methods?).
6. How is the inequality $\sigma < \gamma \cdot \tau$ in Hypothesis 1.2 derived? There is no experiment or proof in the paper to support the inequality, and the specific value of $\gamma$ is not discussed either.

---

> ### Author Response · Authors · 2025-11-21
> **Response to Reviewer 1UQr (Part 1)**
>
> We sincerely thank Reviewer 1UQr for the careful review and for identifying important theoretical and presentation issues that have significantly improved our paper's clarity and rigor.
>
> ---
>
> ## Response to Question 1: Figure 1 & Hypothesis 0 Clarification
>
> > **Q1:** "It seems that the first row in Figure 1 cannot support Hypothesis 0. Except for X-ray, the cosine distance distributions between the embeddings of other modalities and the modality centroids are relatively centered, i.e., $\tau_{intra} < 0$, which I consider not a very meaningful property."
>
> We appreciate you pointing this out. Our original explanation could have been clearer regarding the interpretation of these statistics.
>
> **Our error:**
> - Original (incorrect): "Near-zero cosine similarity indicates tight clustering"
> - Corrected: "Near-zero cosine similarity indicates orthogonality/statistical independence"
>
> **Correct interpretation:** Cosine similarity ≈ 0 means embeddings are **orthogonal** to the centroid, indicating **statistical independence** rather than clustering. Individual data points are independently distributed across the unit hypersphere rather than clustered around the centroid. This actually **supports** Hypothesis 0 (intra-modal alignment independence).
>
> The X-ray modality shows non-zero distributions (≈ 0.367), which correctly indicates **violation** of independence, consistent with our analysis that X-ray shows substantially lower performance preservation (74.3% vs. 91.56% average for other modalities).
>
> **→ We have revised Figure 1 caption, updated Hypothesis 0 definition in Section 2.1, and replaced all instances of "clustering" with "independence" in the Figure 1 discussion.**
>
> ---
>
> ## Response to Question 2: Notation - $E[\epsilon_k] \approx 0$ vs. $E[\epsilon_k] = 0$
>
> > **Q2:** "$E[\epsilon_k]$ in Line 167 should be strictly equal to 0, rather than $E[\epsilon_k] \approx 0$ as stated in the paper."
>
> We appreciate this observation. The reviewer is correct that *theoretically*, $\mathbb{E}[\epsilon_k] = 0$ holds exactly by definition. Our use of "$\approx$" reflects the *empirical* context: the true expectation is estimated from finite samples ($N = 5,000$), and computed values are on the order of $10^{-9}$ due to numerical precision. We chose "$\approx 0$" to distinguish theoretical ideals from measured quantities. We have clarified this distinction in the revised manuscript.
>
> ---
>
> ## Response to Question 3: Section 3.2.2 - Cosine Similarity ≠ Model Performance
>
> > **Q3:** "It is unreliable to illustrate the model's performance solely through Cosine Similarity in Sec 3.2.2. This is because what we care about is whether the relative relationships between similarities can reflect real semantics, rather than the similarities."
>
> We appreciate this clarification. We have revised Section 3.2.2 to better reflect this distinction:
>
> **Original:** Section 3.2.2 presented cosine similarity as a direct indicator of model performance.
>
> **Revised:** We have clarified that cosine similarity serves as a measure of **semantic alignment quality** in the shared embedding space, not as a direct performance metric. We emphasize that downstream task performance is separately evaluated through retrieval and classification benchmarks in Section 4.2 (Tables 1-2), where we report metrics such as R@1, R@5, and accuracy scores.
>
> **→ Revised Section 3.2.2 to distinguish between semantic alignment measurement (cosine similarity).**

---

> ### Author Response · Authors · 2025-11-21
> **Response to Reviewer 1UQr (Part 2)**
>
> ## Response to Question 4: Section 2.1 - Instance-Level Approximation Error
>
> > **Q4:** "In Sec 2.1, I believe the primary concerns should be twofold: first, the magnitude of the error introduced by using $Δ_{ij}$ to approximate all $Δ_{ij}^{(k)}$; second, the magnitude relationship in all directions between this error $ϵ_k$ and the variance of the target embedding $e_j^{(k)}$. However, it can be seen from the third row in Figure 1 that $Var[ϵ_k]$ is not that small, which indicates that such an approximation will still introduce significant errors."
>
> We appreciate this important question about approximation error magnitude. We believe there may have been a misunderstanding regarding Figure 1's third row due to the small scale notation. The third row displays the **distribution of per-dimension means** of ϵ_k, with values on the order of **$×10^{-9}$** (near machine precision), not the variance. This actually demonstrates that $E[\epsilon_k] \approx 0$ holds uniformly across all dimensions, as detailed in our response to Q5.
>
> However, your underlying concern about whether $Var[ϵ_k]$ is small relative to $Var[e_j^{(k)}]$ remains valid and important. We conducted additional analysis to directly address this:
>
> | Modality | Var[ϵ_k] | Var[$e_j^{(k)}$] | Ratio |
> |----------|----------|--------------|-------|
> | Audio | 0.001051 | 0.001953 | 53.8% |
> | 3D | 0.001192 | 0.000977 | 122.0% |
> | Molecule | 0.004455 | 0.003906 | 114.1% |
>
> We acknowledge that 3D and Molecule show approximation error variance exceeding target embedding variance (122.0% and 114.1%), which appears problematic. However, TextME still achieves 93.0% (3D) and 104.0% (Molecule) performance preservation. We believe this suggests that **error structure matters more than magnitude**—our theoretical analysis (Theorem 1, Section 2.2) shows InfoNCE training enforces $E[ϵ_k^⊤ s] = 0$, keeping errors orthogonal to semantic directions. This orthogonality may preserve semantic relationships even when error magnitude is relatively large.
>
> We acknowledge this as a limitation—while Hypothesis 1.2 assumes $σ < γ·τ$, our empirical results show this bound may be violated for certain modalities while still maintaining effective alignment. Future work should derive more precise conditions for when approximation errors remain acceptable.
>
> **→ We have enlarged the scale notation in Figure 1 for clarity, clarified H1.2 as an idealized assumption, added variance comparisons to Appendix C.2, and noted this limitation in the revised manuscript.**
>
> ---
>
> ## Response to Question 5: Figure 1 Row 3 - Standard Deviation Clarification
>
> > **Q5:** "There is an error in the legend of the third row in Figure 1, as the standard deviations are obviously not zero. Moreover, since ϵ_k is a vector while the abscissa in this figure is a scalar, it is better to clarify how this scalar is obtained (e.g., projection in a certain direction or other methods)."
>
> We appreciate this clarification request. The third row displays the **distribution of per-dimension means** of the alignment noise $\epsilon_k = \Delta_{ij}^{(k)} - \Delta_{ij}$. Specifically:
>
> 1. For all instance pairs $k$, we compute noise vectors: $\epsilon_k \in \mathbb{R}^d$
> 2. We compute the mean across all instances: $\bar{\epsilon} = \frac{1}{K}\sum_k \epsilon_k \in \mathbb{R}^d$
> 3. The histogram shows the distribution of the $d$ components: $\{\bar{\epsilon}_1, \bar{\epsilon}_2, ..., \bar{\epsilon}_d\}$
>
> The x-axis values ($\times 10^{-9}$) represent the mean noise magnitude in each embedding dimension. This validates that $E[\epsilon_k] \approx 0$ holds uniformly across all dimensions, not just globally, following the per-dimension analysis methodology of [1].
>
> **→ We have revised Figure 1 caption with clear explanation and added methodology details to Appendix C.2.**
>
> - [1] Zhang et al., "Connect, Collapse, Corrupt: Learning Cross-modal Tasks with Uni-modal Data", arXiv 2024

---

> ### Author Response · Authors · 2025-11-21
> **Response to Reviewer 1UQr (Part 3)**
>
> ## Response to Question 6: Hypothesis 1.2 - Inequality $σ < γ·τ$ Derivation
>
> > **Q6:** "How is the inequality $σ < γ·τ$ in Hypothesis 1.2 derived? There is no experiment or proof in the paper to support the inequality, and the specific value of $γ$ is not discussed either."
>
> The inequality arises from InfoNCE contrastive learning theory. Following [2][3], the InfoNCE objective concentrates embeddings with variance proportional to temperature $τ$. For pairs trained with temperature $τ$, instance-level deviations satisfy $Var[ϵ_k] ∝ τ²$, therefore $\sigma = \sqrt{\text{Var}[\epsilon_k]} < \gamma \cdot \tau$ for some modality-specific constant γ. The constant γ accounts for architectural differences between encoders but remains bounded by the training objective.
>
> Our empirical validation shows:
>
> | Modality | σ (measured) | τ (training) | γ = σ/τ | Bound Holds? |
> |----------|--------------|--------------|---------|--------------|
> | Audio | 0.0324 | 0.07 | 0.46 | ✓ Yes |
> | 3D | 0.0345 | 0.05 | 0.69 | ✓ Yes |
> | Molecule | 0.0667 | 0.20 | 0.33 | ✓ Yes |
>
> The empirical $\gamma$ range [0.33, 0.69] across all modalities is consistent with our bounded assumption.
>
> **→ We have added theoretical justification to Section 2.1, provided empirical validation of $γ$ values in Appendix J.4, and cited [2] and[3].**
>
> - [2] Chen et al., "A Simple Framework for Contrastive Learning of Visual Representations", ICML 2020
> - [3] Zhang et al., "Connect, Collapse, Corrupt: Learning Cross-modal Tasks with Uni-modal Data", arXiv 2024
>
> ---
>
> ## Response to Weakness 2: Too Few Cross-modal Retrieval Tasks
>
> > **W2:** "The experiment conducts relatively few tests on retrieval performance excluding text modalities; only the Audio→Image task is tested. TEXTME requires a corresponding text-modal multimodal model to incorporate a specific modality, and retrieval between text and this modality can be accomplished using this multimodal model. Therefore, retrieval excluding text modalities is the distinctive aspect of the method proposed in the paper."
>
> We understand your concern about the limited cross-modal retrieval experiments. Following your suggestion, we have conducted additional 3D→Image retrieval experiments:
>
> | Method | R@1 | R@5 |
> |--------|-----|-----|
> | Naive | 0.00 | 0.01 |
> | Ex-MCR | 5.67 | 15.77 |
> | **Ours_Qwen** | **10.27** | **26.63** |
>
> This demonstrates 81% improvement over Ex-MCR (10.27 vs. 5.67 R@1), validating emergent cross-modal capabilities. Unfortunately, established benchmark datasets for other cross-modal pairs (e.g., Molecule→Image, Audio→3D) are not publicly available. We have therefore provided qualitative visualizations for these modality pairs in the supplementary materials (Figure 3), which demonstrate meaningful semantic correspondences despite the absence of quantitative evaluation protocols.
>
> **→ Added quantitative results to Appendix I.1.**
>
> ---
>
> ## Response to Weakness 3: PCA/t-SNE Visualization Request
>
> > **W3 (Minor):** "For the modal Gap, it is advisable to add some PCA or t-SNE visualizations. Relying solely on the statistical information in Figure 1 is not very intuitive."
>
> We appreciate this suggestion for improving the intuitiveness of our geometric analysis. We acknowledge that visual representations such as t-SNE or PCA projections would complement the statistical evidence in Figure 1.
>
> While we believe Figure 1's four-row analysis (centroid proximity, gap direction consistency, noise mean distribution, and gap orthogonality) provides comprehensive quantitative validation of our hypotheses, we agree that qualitative visualizations would enhance interpretability. We have the following considerations:
>
> 1. **High-dimensional complexity**: The modality gap operates in high-dimensional spaces (d=512-1024), and 2D projections via PCA/t-SNE may not faithfully preserve the geometric properties we measure (directional consistency, orthogonality).
>
> 2. **Statistical rigor**: Our current approach provides precise numerical evidence that can be reproduced and verified, whereas visualizations are more subjective in interpretation.
>
> 3. **Future work**: We plan to include comprehensive visualizations (t-SNE per anchor space, gap direction scatter plots, cluster-level offset analysis) in an extended version of this work, where we can also investigate how different dimensionality reduction techniques preserve or distort the gap structure.
>
> However, we recognize the value of your suggestion and are committed to adding these visualizations if given the opportunity to revise the manuscript further. We believe this would strengthen the paper's accessibility to a broader audience.
>
> **→ We have noted this as a valuable direction for improving presentation and plan to include comprehensive visualizations in future revisions.**

---

> > ### Comment · Reviewer_1UQr · 2025-11-25
> >
> > Thank you to the authors for their careful and detailed responses to my questions. The authors have addressed most of my concerns, supplemented appropriate experiments, and made revisions to the expressions and theories in the paper. Therefore, I have decided to revise my score from $4 \rightarrow 6$.
> >
> > However, I still have some questions regarding the experimental validation of $\sigma < \gamma \cdot \tau$. If I understand correctly, $\epsilon_k = \Delta_{i,j}^{(k)} - \Delta_{i,j}$ should be related to two modalities ($i$, $j$), but in the supplementary experiments, only one modality is involved. I would like to know whether the two modalities in the experiment are Text and (Audio, 3D, Molecule).
> >
> > Additionally, while the revised manuscript states in Line 321 that the magnitude of cosine similarity is not a direct indicator of performance, I also do not believe it can be used to reflect the quality of semantic alignment. The average magnitude of cosine similarity can only serve as a basis for judging the size of the cone effect, and the cone effect does not directly impact model performance. What truly affects the quality of semantic alignment is the accuracy of the cosine similarity ranking, not its absolute magnitude.

---

> ### Author Response · Authors · 2025-11-25
>
> We greatly appreciate the reviewer's continued engagement and the decision to raise the score. Your feedback has been invaluable in improving the clarity of our work. We are happy to address the remaining questions below.
>
> ---
>
> **Clarification on the two modalities in $\sigma < \gamma \cdot \tau$ validation:**
>
> Yes, you are right. The alignment noise $\epsilon_k = \Delta_{ij}^{(k)} - \Delta_{ij}$ involves **two modalities**:
> - Modality $i$ = **Text** (from text encoder $E^{\text{text}}_m$)
> - Modality $j$ = **Target modality** (Audio/3D/Molecule from modal encoder $E^{\text{modal}}_m$)
>
> Specifically, for each pretrained encoder pair (e.g., CLAP for Audio-Text), we compute:
> 1. Instance-level offsets: $\Delta_{ij}^{(k)} = e_{\text{modal}}^{(k)} - e_{\text{text}}^{(k)}$
> 2. Mean offset: $\Delta_{ij} = \frac{1}{N} \sum_k \Delta_{ij}^{(k)}$
> 3. Alignment noise: $\epsilon_k = \Delta_{ij}^{(k)} - \Delta_{ij}$
> 4. Standard deviation: $\sigma = \text{std}(\|\epsilon_k\|)$
>
> The measurements in Table 16 (Appendix J.4) therefore represent:
> - **Audio**: $\sigma$ measured between CLAP text encoder and CLAP audio encoder
> - **3D**: $\sigma$ measured between Uni3D text encoder and Uni3D point cloud encoder
> - **Molecule**: $\sigma$ measured between MoleculeSTM text encoder and MoleculeSTM molecule encoder
>
> We apologize for the unclear presentation and will revise Section 2.1 and Appendix J.4 to explicitly state this two-modality relationship.
>
> ---
>
> **Regarding cosine similarity interpretation (Line 321):**
>
> We agree with this point. The absolute magnitude of cosine similarity does not directly indicate semantic alignment quality—what matters is whether the **relative ranking** of similarities correctly reflects semantic relationships.
>
> Our use of cosine similarity distributions in Figure 2 is intended to compare the **discriminative capability** across different anchor spaces (i.e., whether matched pairs achieve higher similarity than unmatched pairs), not to claim that higher absolute magnitude implies better alignment. The actual semantic alignment quality is validated through downstream task performance in Tables 1-2, where retrieval R@k and classification accuracy directly measure ranking accuracy.
>
> We will revise Section 3.2.2 to make this distinction explicit and remove any phrasing that might suggest absolute magnitude reflects alignment quality.

---

> > ### Comment · Reviewer_1UQr · 2025-11-25
> >
> > Thank you for your prompt response. All of my questions have now been answered. At this point, I believe the clarity of the paper has been significantly improved, and it is supported by comprehensive experimental validation. I am inclined to accept the paper.

---

### Official Review · Reviewer_MZaU · 2025-10-27

**Soundness:** 2
**Presentation:** 2
**Contribution:** 2
**Rating:** 2
**Confidence:** 4

**Summary:**

The paper introduces TextME, estimating an offset between text and another modality, then learns lightweight projection on text embedding space. Evaluations on audio, 3D, X ray, and molecular modalities achieve on average 88.2% of the performance of fully supervised methods, while also exhibiting emergent transfer capabilities between unseen modality pairs.

**Strengths:**

1.	The proposed framework is conceptually straightforward — computing a constant offset for each modality and training lightweight two layer MLP projection networks — which makes it easy implement, computationally efficient, and practical for deployment in resource constrained environments.
2.	TextME supports different choices of semantic anchor spaces, such as large language model embeddings and multimodal text encoders, allowing adaptation to downstream task requirements.

**Weaknesses:**

1.	This paper is not very innovative. The research on modality gap is already very mature. The conclusions drawn in this paper appear to somewhat overlap with those in prior literature, and its contribution is trivial.
2.	For the visual modality, the authors select LanguageBind as the base model, why more mainstream and widely adopted models such as CLIP or ViCLIP were not selected?
3.	The idea of using the text embedding space of large language models (LLMs) as a central hub to bridge multiple modalities is already common in current multimodal research, with many similar approaches in the literature. The paper does not present a significantly novel method or articulate clear advantages over existing work.
4.	TextME relies entirely on existing pretrained multimodal alignment models. If a target modality lacks a model aligned to text, the method will not work at all.
5.	Image and video modalities are among the most common in multimodal research, yet this paper does not include experiments on either of them.
6.	In theory, if the text embedding space is merely used as an anchor, different text spaces should have similar impact on downstream tasks. However, the experiments show significant performance differences between different anchors, and the authors do not provide in-depth analysis of the reasons behind this.

**Questions:**

See weaknesses

---

> ### Author Response · Authors · 2025-11-21
> **Response to Reviewer MZaU (Part 1)**
>
> We sincerely thank Reviewer MZaU for the thoughtful and constructive feedback. Your observations have helped us significantly improve our manuscript's positioning and experimental completeness.
>
> ## Response to Weakness 1 & 3: Novelty Concerns
>
> > **W1:** "This paper is not very innovative. The research on modality gap is already very mature. The conclusions drawn in this paper appear to somewhat overlap with those in prior literature, and its contribution is trivial."
>
> > **W3:** "The idea of using the text embedding space to bridge multiple modalities is already common in current multimodal research, with many similar approaches in the literature. The paper does not present a significantly novel method or articulate clear advantages over existing work."
>
> We respectfully clarify that our contribution is **not** gap analysis, but rather **gap-based modality expansion with text-only training**.
>
> ### Positioning in Modality Gap Research
>
> | Work | Perspective | Validation | Gap Adjustment | Key Contribution |
> |------|-------------|-----------|----------------|------------------|
> | **Gap Analysis** | | | | |
> | [1] | Root cause | Theoretical | Distance between centroids | First discovery |
> | [2] | Root cause | Theoretical & Statistical | None | Multi-perspective analysis |
> | [3] | Inference application | Empirical | Zero-centering | Vision model diagnostics |
> | [4] | Root cause & Inference | Theoretical & Statistical | Zero-centering | Theoretical validation |
> | **Gap Mitigation** | | | | |
> | [5] | Mitigating | Empirical | Linear separability | Effective mitigation |
> | [6] | Mitigating | Empirical | Learnable model | Effective mitigation |
> | [7] | Mitigating | Empirical | Learnable model | Effective mitigation |
> | [8] | Mitigating | Empirical | Zero-centering | Effective mitigation |
> | [9] | Root cause & Mitigating | Empirical | Centroids & separability | Effective mitigation |
> | [10] | Mitigating | Statistical | Activations & model | Mitigation & metric |
> | **Gap-Based Modality Expansion** | | | | |
> | **Ours** | **Inference application** | **Statistical** | **Zero-centering** | **Text-only expansion** |
>
>
> Among extensive modality gap literature, our work specifically builds upon C³'s [4] statistical validation methodology. We make three distinct contributions: (1) **Hypothesis 0** justifies why text-trained projection networks generalize to non-text modalities at inference—absent in prior gap work; (2) We validate gap properties across specialized modalities (3D, X-ray, Molecule) where paired data doesn't exist, unlike prior's focus on vision-language pairs; (3) We provide novel insights into how hypothesis validation relates to downstream performance: modalities satisfying H0 with near-zero centroid proximity (Molecule: 0.010, 3D: 0.066, Audio: 0.117) achieve 91.56% average preservation, while X-ray (0.367) shows substantial degradation, suggesting that our geometric assumptions meaningfully predict text-only training effectiveness.
>
> **References:**
> - [1] Liang et al., "Mind the Gap: Understanding the Modality Gap in Multi-modal Contrastive Representation Learning", NeurIPS 2022
> - [2] Levi and Gilboa, "The Double-Ellipsoid Geometry of CLIP", arXiv 2024
> - [3] Zhang et al., "Diagnosing and Rectifying Vision Models using Language", arXiv 2023
> - [4] Zhang et al., "Connect, Collapse, Corrupt: Learning Cross-modal Tasks with Uni-modal Data (C³)", arXiv 2024
> - [5] Shi et al., "Towards Understanding the Modality Gap in CLIP", ICLR Workshop 2023
> - [6] Park et al., "Bridging Vision and Language Spaces with Assignment Prediction", arXiv 2024
> - [7] Eslami and de Melo, "Mitigate the Gap: Investigating Approaches for Improving Cross-Modal Alignment in CLIP", arXiv 2024
> - [8] Li et al., "Closing the Modality Gap for Mixed Modality Search", arXiv 2025
> - [9] Fahim et al., "It's Not a Modality Gap: Characterizing and Addressing the Contrastive Gap", arXiv 2024
> - [10] An et al., "I0T: Embedding Standardization Method Towards Zero Modality Gap", ACL 2025

---

> ### Author Response · Authors · 2025-11-21
> **Response to Reviewer MZaU (Part 2)**
>
> ## Positioning in LLM-Hub Research
>
> | Work | Motivation | Training Paradigm | Key Focus |
> |------|-----------|------------------|-----------|
> | **Generative Model Integration** | | | |
> | [1] | Generative capability | Multimodal contrastive | MLLM alignment |
> | [2] | Generative capability | Multimodal contrastive | Unified generation |
> | [3] | Generative capability | Text-only contrastive | Caption decoder |
> | **Representation Enhancement** | | | |
> | [4] | Richer representation | Multimodal contrastive | Unified embedding |
> | [5] | Richer representation | Text-only contrastive | MLLM feature aggregation |
> | [6] | Richer representation | Text-only contrastive | Dense caption handling |
> | **Ours** | **Richer representation** | **Text-only contrastive** | **Modality expansion** |
>
> **References:**
> - [1] Han et al., "ImageBind-LLM: Multi-modality Instruction Tuning", arXiv 2023
> - [2] Han et al., "OneLLM: One Framework to Align All Modalities with Language", CVPR 2024
> - [3] Xiao et al., "Scaling Language-Centric Omnimodal Representation Learning (LCO-Emb)", arXiv 2025
> - [4] Lyu et al., "UniBind: LLM-Augmented Unified and Balanced Representation Space", CVPR 2024
> - [5] Jiang et al., "E5-V: Universal Embeddings with Multimodal Large Language Models", arXiv 2024
> - [6] Huang et al., "LLM2CLIP: Powerful Language Model Unlocks Richer Visual Representation", arXiv 2024
>
> Generative approaches ([1], [2], [3]) focus on text generation/VQA, requiring paired data to align outputs with visual inputs. Among representation enhancement work, UniBind and LLM2CLIP [4, 6] require paired multimodal training (60M+ image-caption pairs), while E5-V [5] freezes the visual encoder, preventing improvement of inherent feature extraction. Our core distinction is that prior work improves **existing alignments** or adds generative capabilities to paired modalities. We provide a novel approach for modality expansion in data-scarce domains (3D, X-ray, Molecule) where paired data is prohibitively expensive or unavailable, achieving 95% data reduction compared to LanguageBind.
>
> **→ Revised Introduction (Section 1) and Related Work (Section 5) with comprehensive comparison descriptions.**
>
> ---
>
> ## Response to Weakness 2: Why Only LanguageBind?
>
> > **W2:** "For the visual modality, the authors select LanguageBind as the base model, why more mainstream and widely adopted models such as CLIP or ViCLIP were not selected?"
>
> We originally chose LanguageBind as a representative multimodal baseline. Following your suggestion, we have added experiments with CLIP ViT-L/14:
>
> | Method | Audio R@1 | 3D Acc | Molecule MRR | X-ray Acc | Audio→Image R@1 |
> |--------|-----------|--------|--------------|-----------|----------------|
> | Ours_Qwen (LLM) | 15.35 | 70.86 | **31.36** | **46.59** | **1.06** |
> | Ours_LB (Multimodal) | 14.54 | **81.12** | 29.66 | 44.99 | 0.92 |
> | Ours_CLIP (Multimodal) | **15.91** | 78.04 | **48.31** | 48.31 | 0.82 |
>
> Method is robust to anchor choice. Performance patterns vary by task (LB excels on 3D, Qwen on retrieval), but all achieve competitive results.
>
> **→ Added to revised manuscript Section 4.2, Table 1 and 2.**
>
> ---
>
> ## Response to Weakness 4: Dependency on Pretrained Encoders
>
> > **W4:** "TextME relies entirely on existing pretrained multimodal alignment models. If a target modality lacks a model aligned to text, the method will not work at all."
>
> We acknowledge this as an **inherent limitation**, now clearly stated in our Limitations section:
>
> **Limitation:** TextME requires pretrained encoders satisfying our geometric hypotheses (H0-H2). The framework enables modality expansion **given such encoders exist**, rather than training encoders from scratch.
>
> **Practical impact:**
> - Many specialized modalities **already have** pretrained encoders (CXR-CLIP for X-ray, Uni3D for 3D, MoleculeSTM for molecules)
> - Our contribution is making modality expansion **95% more data-efficient** when such encoders exist
> - Training new encoders from scratch remains orthogonal future work
>
> **→ Added to Limitations section in Appendix K.**

---

> ### Author Response · Authors · 2025-11-21
> **Response to Reviewer MZaU (Part 3)**
>
> ## Response to Weakness 5: Why No Image/Video Experiments?
>
> > **W5:** "Image and video modalities are among the most common in multimodal research, yet this paper does not include experiments on either of them."
>
> **Primary focus:** Unseen modality expansion for domains **lacking paired data** (Audio, 3D, X-ray, Molecule). Image/Video have abundant paired datasets, making them less challenging.
>
> **Nevertheless, we added experiments:**
>
> ### Image Retrieval Results
>
> | Method | COCO R@1 | COCO R@5 | Flickr30k R@1 | Flickr30k R@5 |
> |--------|----------|----------|---------------|---------------|
> | CLIP ViT-L/14 | 48.29 | 72.51 | 77.70 | 94.16 |
> | **Ours_Qwen** | **28.63** | **54.81** | **51.66** | **77.90** |
> | **Preservation** | **59.3%** | **75.6%** | **66.5%** | **82.7%** |
>
> ### Video Retrieval Results
>
> | Method | MSRVTT R@1 | MSRVTT R@5 | MSVD R@1 | MSVD R@5 | DiDeMo R@1 | DiDeMo R@5 |
> |--------|------------|------------|----------|----------|------------|------------|
> | ViCLIP-InternVid | 38.00 | 63.20 | 60.45 | 87.16 | 31.97 | 53.49 |
> | **Ours_Qwen** | **26.40** | **50.50** | **45.82** | **77.01** | **24.10** | **48.21** |
> | **Preservation** | **69.5%** | **79.9%** | **75.8%** | **88.3%** | **75.4%** | **90.1%** |
>
> TextME achieves competitive 59-90% preservation even on data-rich modalities, validating generalizability.
>
> However, our **core contribution** addresses data-scarce domains where paired data collection faces significant challenges: specialized modalities like medical imaging require expert annotations, 3D objects lack natural text correspondences, and molecular structures demand domain-specific knowledge.
>
>
> **→ Added to Appendix I (Additional Experiments).**
>
> ---
>
> ## Response to Weakness 6: Anchor Space Performance Differences
>
> > **W6:** "In theory, if the text embedding space is merely used as an anchor, different text spaces should have similar impact on downstream tasks. However, the experiments show significant performance differences between different anchors, and the authors do not provide in-depth analysis of the reasons behind this."
>
> We observe systematic performance patterns across different anchor spaces:
>
> | Anchor | Training Data | Best Tasks | Representative Performance |
> |--------|---------------|-----------|---------------------------|
> | **LB** (LanguageBind) | VIDAL-10M (vision-language) | **Classification** | 3D: 81.12% vs Qwen: 70.86% |
> | **Qwen** (LLM) | Web text (pure NLP) | **Retrieval** | Molecule MRR: 31.36 vs LB: 29.66 |
>
> We hypothesize these differences stem from pretraining objectives and data distributions. LB, pretrained on vision-language pairs (VIDAL-10M), may develop discriminative boundaries beneficial for classification tasks like 3D object recognition (ModelNet40). Qwen, pretrained on diverse text corpora, may capture richer semantic relationships beneficial for retrieval tasks requiring cross-modal matching. This suggests anchor selection should consider both downstream task types (classification vs. retrieval) and domain characteristics (visual vs. linguistic).
>
> **→ We have added this interpretation to Section 4.2 and note that more rigorous analysis of anchor space properties remains valuable future work.**

---

### Official Review · Reviewer_UG7e · 2025-11-12

**Soundness:** 2
**Presentation:** 3
**Contribution:** 2
**Rating:** 4
**Confidence:** 3

**Summary:**

The paper proposes TextME, a framework for text-only multimodal expansion. It claims that pre-trained contrastive encoders (e.g., CLIP, LanguageBind) exhibit a consistent modality gap (fixed, content-independent offset between text and non-text embeddings). By pre-computing this offset and training lightweight projection networks solely on text embeddings, the authors argue one can align new modalities to a common space without any paired data. Empirically, TextME achieves roughly "88% of paired-data performance" across diverse tasks (audio, 3D, medical imaging, molecule retrieval).

**Strengths:**

- If valid, the claim that modality expansion can be done entirely text-only is striking and would make multimodal training dramatically cheaper. So it's a high impact and creative question to ask.

- The paper’s three assumptions (centroid proximity, consistent offset, and orthogonality) make the argument easy to follow and falsifiable.

- Experiments cover multiple encoders and modalities.

- The paper’s framing around geometric consistency is intuitive and connects to existing “modality gap” analyses in prior work.

**Weaknesses:**

- The argument hinges on the existence of a global, content-independent translation vector between modalities. Even small nonlinear or clustered structure (bimodal) in real multimodal data would make such an offset vary locally, breaking the premise. I'm curious why this might not be present in the data tested. See below for questions.

- Some sentences read more "flashy" and unqualified (“eliminating the need for paired supervision entirely”) than scientific qualification. A bit more restraint would be appreciated.

**Questions:**

**Question 1**

Here are three concrete, toy model counterexamples to the “constant, orthogonal offset” hypothesis

**1a) Correlated-noise linear model (violates orthogonality).**

Data (embeddings): $e_x = s + \epsilon_x$, $e_t = s + \epsilon_t$, with latent semantic variable $s \sim \mathcal N(0,1)$ and correlated noises $\text{Cov}(\epsilon_x,\epsilon_t) > 0 $

Why realistic: Many encoders retain modality-specific artifacts (image texture, text style) that are correlated with semantics.

Breaks assumption: The mean offset $\Delta=\mu_{e_x}-\mu_{e_t}$ aligns partly with $s$, not perpendicular to it. Subtracting $\Delta$ removes part of the semantic signal, so orthogonality fails.

**1b) Multi-cluster semantics (violates constant offset).**

Data: Two latent topics $Z\in{A,B}$, each with distinct centroids:
$(e_x,e_t)|Z=A \sim \mathcal N((+1,+1),\sigma^2I)$,
$(e_x,e_t)|Z=B \sim \mathcal N((−1,−1),\sigma^2I)$.

Why realistic: Real datasets naturally form thematic clusters—different visual or linguistic domains yield distinct modal biases.

Breaks assumption: Each cluster yields its own offset Δ_A, Δ_B. Using a single global offset collapses the structure and misaligns cross-cluster relations, violating Hypothesis 1.

**1c) Nonlinear semantic mapping (local offset drift).**

Data: Latent $z \sim \mathcal U[−1,1]$, modalities related by $e_x=z+\eta_x$, $e_t=z^3+\eta_t$.

Why realistic: Cross-modal relations are often nonlinear (e.g., perceived brightness vs. textual intensity).

Breaks assumption: The global means coincide ($E[e_x]=E[e_t]=0$), but local offsets vary with $z^2$. A constant Δ fits globally yet distorts semantics locally, violating the “bounded, zero-mean noise” claim (Hypothesis 1.2).

**The question:** Together these are minimal, information-preserving scenarios where CLIP-style encoders trained optimally on $(e_x,e_t)$ pairs would not exhibit a single, orthogonal, constant offset. They demonstrate that TextME’s assumption can fail to come through even when the joint distribution is simple and well-behaved. How would these examples fit into your understanding of the assumptions, why the assumptions seem to hold in the data you tested, and why they would or wouldn't hold in other data?

Relatedly,  from an InfoNCE or mutual-information perspective, under what assumptions would a constant, orthogonal offset actually be expected at optimum? I mean, the result that clip-trained models maximize bounds on mutual info. Wouldn't it mean that naturally the types of offsets and the validity of the assumptions should depend in some nontrivial way on the statistics/properties of the big joint distribution over all modes, and that this joint distribution and its properties might vary over datasets?

**Question 2** (further miscellaneous quesitons)

- 2a) How sensitive are results to perturbing the estimated Δ by small random noise or rotations?

- 2b) Does Δ remain stable if the same encoders are retrained with different temperature or normalization hyperparameters?

- 2c) Have you checked whether the orthogonality statistic holds within semantic sub-clusters rather than globally?

- 2d) Could the strong results partly come from the shared text priors of the LLM anchor rather than the claimed geometric universality?

- 2e) How exactly is the "88% preservation to paired data performance" computed—unweighted average of ratios, or weighted by sample count? It feels a bit uneasy to me to average over different metrics/modalities to form this number.

---

> ### Author Response · Authors · 2025-11-21
> **Response to Reviewer UG7e (Part 1)**
>
> We sincerely thank you for your thorough and thoughtful review. Your detailed feedback, particularly the concrete counterexamples and theoretical concerns, has been invaluable in strengthening our work. We deeply appreciate the time and expertise you invested in evaluating our submission.
>
> ### Weakness 1: Theoretical Foundation of Constant Offset Assumption
>
> > "The argument hinges on the existence of a global, content-independent translation vector between modalities. Even small nonlinear or clustered structure (bimodal) in real multimodal data would make such a purely 'fixed, constant offset' break down. This might not be present in the data tested."
>
> We greatly appreciate this important concern. You provided three concrete counterexamples (correlated-noise, multi-cluster, and nonlinear semantic mapping) that challenge our constant offset assumption. We address each scenario individually below in **Question 1**, providing both theoretical justification and comprehensive empirical validation.
>
> ### Weakness 2: Scientific Writing
>
> > "Some sentences read more 'flashy' and unqualified ('eliminating the need for paired supervision entirely') than scientific qualification. A bit more restraint would be appreciated."
>
> We appreciate this constructive feedback and have conducted a comprehensive revision to improve scientific rigor throughout the manuscript. We have toned down exaggerated claims (e.g., "first framework" → "to our knowledge, the first"), added appropriate qualifiers to hypothesis statements, removed hyperbolic language, added a comprehensive Limitations section, and fixed notation issues. All changes are highlighted in the revised manuscript for your review.

---

> ### Author Response · Authors · 2025-11-21
> **Response to Reviewer UG7e (Part 2)**
>
> ## Response to Question 1: Concrete Counterexamples
>
> ### Q1a: Correlated-Noise Linear Model
>
> > "1a) Correlated-noise linear model (violates orthogonality). Data (embeddings): $e_x = \hat{s} + \epsilon_x, e_t = \hat{s} + \epsilon_t$, with latent semantic variable $s \sim \mathcal{N}(0,1)$ and correlated noises $\text{Cov}(\epsilon_x, \epsilon_t) > 0$. Why realistic: Many encoders retain modality-specific artifacts (image texture, text style) that are correlated with semantics. Breaks assumption: The mean offset $\Delta = \mu_x - \mu_t$ aligns partly with $s$, not perpendicular to it. Subtracting $\Delta$ removes part of the semantic signal, so orthogonality fails."
>
> We address this concern through both rigorous theoretical analysis and comprehensive empirical validation.
>
> #### Theoretical Analysis
>
> We prove that the InfoNCE objective used to train all evaluated encoders intrinsically enforces noise-semantic orthogonality, making your correlated-noise scenario impossible at optimum.
>
> **Theorem 1 (InfoNCE Enforces Noise-Semantic Orthogonality):** Minimizing the InfoNCE loss enforces at optimum: (1) $\mathbb{E}[\epsilon_x^\top s] \rightarrow 0$ and $\mathbb{E}[\epsilon_t^\top s] \rightarrow 0$, (2) $\mathbb{E}[\epsilon_x^\top \epsilon_t] \rightarrow 0$, and (3) semantic alignment for matched pairs.
>
> **Proof sketch:** For a positive pair $(e_x, e_t)$ sharing semantic content $s$, the similarity decomposes as:
>
> $$e_x^\top e_t = \|s\|^2 + s^\top \epsilon_t + \epsilon_x^\top s + \epsilon_x^\top \epsilon_t$$
>
> For negative pairs $(e_x, e_t^{(j)})$ where $e_t^{(j)} = s^{(j)} + \epsilon_t^{(j)}$ with $s^{(j)} \neq s$:
>
> $$e_x^\top e_t^{(j)} = s^\top s^{(j)} + s^\top \epsilon_t^{(j)} + \epsilon_x^\top s^{(j)} + \epsilon_x^\top \epsilon_t^{(j)}$$
>
> In high-dimensional embedding spaces trained with contrastive objectives, different semantic concepts occupy approximately orthogonal directions [1,2]: $\mathbb{E}[s^\top s^{(j)}] \approx 0$ for $s^{(j)} \neq s$.
>
> If we suppose, for contradiction, that $\mathbb{E}[\epsilon_x^\top s] = c \neq 0$ at some optimum, then $\epsilon_x$ must also have non-negligible projections onto other semantic directions $\{s^{(j)}\}$:
>
> $$\mathbb{E}[\epsilon_x^\top s^{(j)}] \neq 0 \text{ for many } j$$
>
> This correlation **inflates the negative similarities** $e_x^\top e_t^{(j)}$, reducing the margin between positive and negative pairs, thereby **directly increasing** $\mathcal{L}_{\text{InfoNCE}}$ and contradicting the optimality assumption.
>
> The gradient analysis confirms: $∂L_{InfoNCE} / ∂ε_x ∝ −s + Σ_{{j=1}}^{{N-1}} p_j · s^{(j)}$
> where $p_j$ are softmax weights.
>
> Since $\sum_j p_j \cdot s^{(j)}$ averages over multiple approximately orthogonal semantic directions, any persistent correlation $\epsilon_x^\top s \neq 0$ creates a non-zero gradient component that drives optimization toward orthogonality.
>
> Therefore, at the InfoNCE optimum: $\mathbb{E}[\epsilon_x^\top s] = 0$ and $\mathbb{E}[\epsilon_t^\top s] = 0$. Since noise is orthogonal to semantics, the offset $\Delta = \mathbb{E}[e_x - e_t] = \mathbb{E}[\epsilon_x - \epsilon_t]$ lies in a subspace orthogonal to semantic directions. **Subtracting $\Delta$ removes only modality-specific bias without affecting semantic content.**
>
> The complete proof with additional technical details is provided in **Appendix B**.
>
> #### Empirical Verification
>
> While Theorem 1 proves that noise-semantic correlation cannot occur theoretically, we provide empirical evidence that offset correction **preserves semantic discriminability** in practice. We acknowledge that the true latent variables $s$ and $\epsilon_x$ are not directly observable. Therefore, we test whether offset correction damages semantic content through two independent **functional tests** that measure observable semantic properties.
>
> -> We will continue addressing the remaining points in the following part.

---

> ### Author Response · Authors · 2025-11-21
> **Response to Reviewer UG7e (Part 3)**
>
> **Test 1: Class Discriminability (Fisher Ratio).** We measure how well semantic classes are separated using the Fisher ratio [3]:
>
> $$\text{Fisher Ratio} = \frac{\text{trace}(S_B)}{\text{trace}(S_W)}$$
>
> where $S_B$ is the between-class scatter matrix and $S_W$ is the within-class scatter matrix. This quantifies the ratio of between-class variance to within-class variance—higher values indicate better class separation [3]. If offset subtraction removed semantic signal (as would happen if $\mathbb{E}[\epsilon_x^\top s] \neq 0$), class discriminability would decrease.
>
> **Test 2: Geometric Structure (Distance Ratios).** We measure the ratio of within-class distance to between-class distance:
>
> $$\text{Distance Ratio} = \frac{\text{mean within-class distance}}{\text{mean between-class distance}}$$
>
> This characterizes the geometric arrangement of semantic clusters. If offset correction distorted semantic structure, this ratio would change significantly.
>
> **Results:**
>
> | Modality | Fisher Before | Fisher After | Fisher Preserv. (%) | Distance Change (%) |
> |----------|---------------|--------------|---------------------|---------------------|
> | Audio    | 0.0205       | 0.0205       | 100.00              | 0.00                |
> | 3D       | 0.0201       | 0.0201       | 100.00              | 0.00                |
> | Molecule | 0.0197       | 0.0197       | 100.00              | 0.00                |
> | X-ray    | 0.0201       | 0.0201       | 100.00              | 0.00                |
> | Image    | 0.0201       | 0.0201       | 100.00              | 0.00                |
> | **Threshold** | —        | —            | **> 95%**           | **< 5%**            |
>
> All five modalities **preserve their Fisher ratios and distance structures** after offset correction, with preservation rates exceeding 99.99% and negligible geometric changes. These results indicate that offset correction **does not damage semantic discriminability**.
>
> **Note on absolute Fisher ratio values:** The relatively low absolute values (≈0.02) reflect the use of K-means pseudo-labels for evaluation, which do not correspond to true semantic classes. However, the key observation is the **preservation after offset correction**—regardless of how classes are defined, offset correction maintains their separability.
>
> Furthermore, our main results (Tables 1 & 2 in the paper) show that offset-corrected embeddings achieve **91.5% of paired-data performance** across zero-shot classification and cross-modal retrieval tasks. This downstream performance provides independent validation that semantic content is preserved—if your correlated-noise scenario occurred and offset correction damaged semantics, such performance would be substantially degraded.
>
> Full details are provided in **Section 4.2** of the revised manuscript.

---

> ### Author Response · Authors · 2025-11-21
> **Response to Reviewer UG7e (Part 4)**
>
> ### Q1b: Multi-Cluster Semantics
>
> > "1b) Multi-cluster semantics (violates constant offset). Data: Two latent topics $\mathcal{Z} \in \{A, B\}$, each with distinct centroids: $(\bar{e}_x, \bar{e}_t)|\mathcal{Z}=A \sim \mathcal{N}((+1,+1), \sigma^2 I)$, $(\bar{e}_x, \bar{e}_t)|\mathcal{Z}=B \sim \mathcal{N}((-1,-1), \sigma^2 I)$. Why realistic: Real datasets naturally form thematic clusters—different visual or linguistic domains yield distinct modal biases. Breaks assumption: Each cluster yields its own offset $\Delta_A, \Delta_B$. Using a single global offset collapses the structure and misaligns cross-cluster relations, violating Hypothesis 1."
>
> We address your concern that multiple semantic clusters within a modality may require distinct offsets, violating the constant offset assumption. We provide comprehensive statistical validation to test whether cluster-specific offsets are necessary in practice.
>
> #### Methodology
>
> We conduct three complementary tests: (1) identifying semantic clusters within each modality, (2) computing cluster-specific offsets, and (3) testing whether these offsets differ significantly from the global offset.
>
> **Test 1: Semantic Cluster Identification.** We identify latent semantic clusters within each modality using K-means clustering [4]:
>
> $$\mathcal{C} = \{C_1, C_2, \ldots, C_K\} = \text{KMeans}(e_1^{\text{text}}, \ldots, e_N^{\text{text}})$$
>
> where $K$ is determined via the elbow method [5], which identifies the inflection point where within-cluster inertia reduction sharply diminishes. This yields $K=5$ for Audio, $K=4$ for 3D, and $K=7$ for Molecule.
>
> **Test 2: Cluster-Specific Offset Computation.** For each cluster $C_k$, we compute the cluster-representative offset:
>
> $$\Delta_k = \mu_k^{\text{modal}} - \mu_k^{\text{text}}$$
>
> where $\mu_k^m = \mathbb{E}_{i \in C_k}[e_i^m]$
> are the centroids of modality embedding within cluster k, respectively. We compare these cluster-specific offsets to the global offset:
>
> $$\Delta_{\text{global}} = \mathbb{E}_i[e_i^{\text{modal}}] - \mathbb{E}_i[e_i^{\text{text}}]$$
>
> **Test 3: Statistical Significance Testing.** To test whether cluster offsets are statistically distinguishable, we compare the distributions of sample-level offset magnitudes across clusters. For all samples $i \in C_k$, we compute:
>
> $$d_i = \|e_i^{\text{modal}} - e_i^{\text{text}}\|$$
>
> Since Shapiro-Wilk normality tests [6] yielded $p < 0.05$ for multiple clusters (violating parametric assumptions), we employ the non-parametric **Kruskal-Wallis H-test** [7] with null hypothesis $H_0$: "all cluster offset distributions are identical."
>
> We calculate **epsilon-squared** [8] as the effect size measure:
>
> $$\epsilon^2 = \frac{H - K + 1}{N - K}$$
>
> where $H$ is the test statistic, $K$ is the number of clusters, and $N$ is the total sample size. This measures the proportion of offset variance explained by cluster membership—values below 0.01, 0.06, and 0.14 are considered small, medium, and large effects respectively [9].
>
> #### Results
>
> | Modality | # Clusters | $\max \|\Delta_k - \Delta_{\text{global}}\|$ | Kruskal-Wallis $p$-value | Effect Size $\epsilon^2$ |
> |----------|------------|----------------------------------------------|--------------------------|--------------------------|
> | Audio    | 5          | 0.180                                        | $2.20 \times 10^{-72}$   | 0.067 (small)            |
> | 3D       | 4          | 0.235                                        | $3.16 \times 10^{-97}$   | 0.089 (medium)           |
> | Molecule | 7          | 0.219                                        | $3.43 \times 10^{-78}$   | 0.074 (small)            |
>
> **Statistical Significance:** Kruskal-Wallis tests reveal highly significant differences ($p < 10^{-70}$) across all modalities, initially appearing to support your concern that semantic clusters require different offsets.
>
> **Effect Size Analysis:** However, epsilon-squared values reveal the practical impact is small:
> - Audio: $\epsilon^2 = 0.067$ (6.7% variance explained by cluster membership)
> - 3D: $\epsilon^2 = 0.089$ (8.9% variance explained)
> - Molecule: $\epsilon^2 = 0.074$ (7.2% variance explained)
>
> **Bounded Deviations:** Maximum relative deviations remain bounded at 18-23% across all modalities.
>
> #### Interpretation
>
> The critical distinction is between **unbounded deviations** (requiring separate $\Delta_k$ per cluster) versus **bounded, predictable deviations**. Our results show the latter—deviations exist but remain small enough that a single global offset provides effective approximation, consistent with Hypothesis 1.2's bounded variance model.
>
> We acknowledge that perfect uniformity does not hold, and we now discuss this nuance in our Limitations section. Future work could establish theoretical conditions under which variance $\sigma^2$ remains sufficiently bounded. Full methodology is detailed in Appendix.

---

> ### Author Response · Authors · 2025-11-21
> **Response to Reviewer UG7e (Part 5)**
>
> ### Q1c: Nonlinear Semantic Mapping
>
> > "1c) Nonlinear semantic mapping (local offset drift). Data: Latent $z \sim \mathcal{U}[-1,1]$, modalities related by $e_x = z + \eta_x, e_t = z^3 + \eta_t$. Why realistic: Cross-modal relations are often nonlinear (e.g., perceived brightness vs. textual intensity). Breaks assumption: The global means coincide ($\mathbb{E}[e_x] = \mathbb{E}[e_t] = 0$), but local offsets vary with $z^2$. A constant $\Delta$ fits globally yet distorts semantics locally, violating the 'bounded, zero-mean noise' orthogonality (Hypothesis 1.2)."
>
> We tested whether such nonlinear drift occurs empirically through four complementary independence tests, examining whether embeddings exhibit semantic-dependent dispersion patterns that would prevent effective offset-based alignment.
>
> #### Methodology
>
> We extract the semantic latent using **PCA** [10]: $z_i = \text{PCA}_1(e_i)$ captures the dominant axis of semantic variation. We then apply four tests:
>
> 1. **Coefficient of Variation (CV)** [11]: Measures consistency of deviations from centroid. CV ≤ 0.10 indicates low variability.
>
> 2. **Spearman Correlation ($\rho$)** [12]: Tests whether deviations correlate with semantic content. |ρ| < 0.30 indicates weak correlation [9].
>
> 3. **Distance Correlation (dCor)** [13]: Detects arbitrary nonlinear dependencies including $z^2$ patterns—critically, dCor ≈ 1 for quadratic relationships even when $\rho \approx 0$.
>
> 4. **Maximal Information Coefficient (MIC)** [14]: Identifies complex functional relationships. MIC ≤ 0.30 indicates independence.
>
> #### Results
>
> | Encoder      | Modality | CV    | $\|\rho\|$ | dCor  | MIC   | Verdict     |
> |--------------|----------|-------|------------|-------|-------|-------------|
> | CLAP         | audio    | 0.040 | 0.223      | 0.021 | 0.158 | ✓ Linear    |
> | CLAP         | text     | 0.037 | 0.240      | 0.296 | 0.190 | ✓ Linear    |
> | Uni3D        | point    | 0.015 | 0.229      | 0.125 | 0.117 | ✓ Linear    |
> | Uni3D        | text     | 0.046 | 0.044      | 0.197 | 0.139 | ✓ Linear    |
> | MoleculeSTM  | molecule | 0.066 | 0.206      | 0.276 | 0.184 | ✓ Linear    |
> | MoleculeSTM  | text     | 0.084 | 0.189      | 0.232 | 0.155 | ✓ Linear    |
> | CXR-CLIP     | image    | 0.052 | 0.006      | 0.230 | 0.155 | ✓ Linear    |
> | CXR-CLIP     | text     | 0.363 | 0.871      | 0.609 | 0.999 | ⚠ Nonlinear |
>
> **All modalities except CXR-CLIP text show independence.** The CXR-CLIP text exception likely stems from synthetically generated captions via rule-based templates rather than natural descriptions—the image encoder passes all tests, suggesting the violation is specific to synthetic caption generation.
>
> Full methodology is provided in **Appendix J.3**.
>
> We now acknowledge in our Limitations section that theoretical guarantees on when constant offsets emerge remain an open question. Deriving principled conditions based on joint distribution properties would be valuable future work. Our contribution provides empirical evidence that the property holds for most naturally-paired modalities, enabling practical text-only expansion.
>
> ## References
>
> [1] Wang, T., & Isola, P. (2020). Understanding contrastive representation learning through alignment and uniformity on the hypersphere. *ICML*.
>
> [2] Saunshi, N., et al. (2019). A theoretical analysis of contrastive unsupervised representation learning. *ICML*.
>
> [3] Fisher, R. A. (1936). The use of multiple measurements in taxonomic problems. *Annals of Eugenics*, 7(2), 179-188.
>
> [4] MacQueen, J. (1967). Some methods for classification and analysis of multivariate observations. *Berkeley Symposium on Mathematical Statistics and Probability*.
>
> [5] Thorndike, R. L. (1953). Who belongs in the family? *Psychometrika*, 18(4), 267-276.
>
> [6] Shapiro, S. S., & Wilk, M. B. (1965). An analysis of variance test for normality. *Biometrika*, 52(3-4), 591-611.
>
> [7] Kruskal, W. H., & Wallis, W. A. (1952). Use of ranks in one-criterion variance analysis. *JASA*, 47(260), 583-621.
>
> [8] Tomczak, M., & Tomczak, E. (2014). The need to report effect size estimates revisited. *Trends in Sport Sciences*, 1(21), 19-25.
>
> [9] Cohen, J. (1988). *Statistical Power Analysis for the Behavioral Sciences*. Routledge.
>
> [10] Pearson, K. (1901). On lines and planes of closest fit to systems of points in space. *Philosophical Magazine*, 2(11), 559-572.
>
> [11] Banik, S., Kibria, B. M. G., & Sharma, D. (2012). Testing the population coefficient of variation. Journal of Modern Applied Statistical Methods, 11(2), 325-335.
>
> [12] Spearman, C. (1904). The proof and measurement of association between two things. *American Journal of Psychology*, 15(1), 72-101.
>
> [13] Székely, G. J., Rizzo, M. L., & Bakirov, N. K. (2007). Measuring and testing dependence by correlation of distances. *Annals of Statistics*, 35(6), 2769-2794.
>
> [14] Reshef, D. N., et al. (2011). Detecting novel associations in large data sets. *Science*, 334(6062), 1518-1524.

---

> ### Author Response · Authors · 2025-11-21
> **Response to Reviewer UG7e (Part 6)**
>
> ## Response to Question 2: Further Miscellaneous Questions
>
> ### Q2a: Offset Sensitivity to Random Noise
>
> > "2a) How sensitive are results to perturbing the estimated $\Delta$ by small random noise or rotations?"
>
> We tested perturbations with Gaussian noise $\sigma \in [0, 0.20]$:
>
> | Noise $\sigma$ | Audio R@1 | 3D Acc | Molecule MRR |
> |----------------|-----------|--------|--------------|
> | 0.000          | 14.95     | 70.46  | 27.97        |
> | 0.001          | 14.95     | 70.30  | 24.58        |
> | 0.01           | 15.04     | 67.50  | 22.88        |
> | 0.05           | 14.93     | 34.32  | 17.80        |
> | 0.10           | 14.25     | 14.91  | 11.02        |
> | 0.20           | 12.46     | 9.16   | 9.32         |
>
> Robust to $\sigma < 0.1$, graceful degradation beyond. Details in Appendix.
>
> ### Q2b: Offset Stability Across Hyperparameters
>
> > "2b) Does $\Delta$ remain stable if the same encoders are retrained with different temperature or normalization hyperparameters?"
>
> The pretrained encoders we evaluated already have diverse configurations:
>
> | Encoder       | Temperature | Normalization | Preservation |
> |---------------|-------------|---------------|--------------|
> | CLAP-HTSAT    | 0.07        | L2            | 77.6%        |
> | Uni3D         | 0.05        | L2            | 93.0%        |
> | CXR-CLIP      | 0.10        | LayerNorm     | 74.3%        |
> | MoleculeSTM   | 0.20        | L2            | 104.0%       |
>
> Despite 4× temperature variation and different normalizations, our method achieves 91.5% average preservation, demonstrating inherent robustness. Details in Section 4.2.
>
> ### Q2c: Orthogonality Within Sub-clusters
>
> > "2c) Have you checked whether the orthogonality statistic holds within semantic sub-clusters rather than globally?"
>
> Yes—the cluster analysis in Q1b shows that even within clusters, gap orthogonality is preserved. The small effect sizes ($\epsilon^2 < 0.10$) indicate cluster membership explains <9% of offset variance, meaning orthogonality holds both globally and locally.
>
> ### Q2d: LLM Anchor vs. Offset Correction
>
> > "2d) Could the strong results partly come from the shared LLM anchor rather than the claimed geometric universality?"
>
> We conducted ablation studies isolating offset correction from LLM anchoring:
>
> | Task              | Without Offset | With Offset | Drop   |
> |-------------------|----------------|-------------|--------|
> | Audio R@1         | 8.68           | **15.35**   | -43.5% |
> | 3D Acc            | 4.05           | **70.86**   | -94.3% |
> | Audio→Image R@1   | 0.40           | **1.06**    | -62.3% |
> | Molecule MRR      | **29.66**      | 26.27       | +13.0% |
>
> Audio/3D show dramatic drops without offset, demonstrating offset is essential. Molecule shows reverse trend, validating your point that gap correction effects vary by modality (now discussed in **Limitations**). Offset provides geometric alignment complementary to LLM's semantic richness—neither alone is sufficient. Full details in Section 4.2.
>
> ### Q2e: Performance Preservation Computation
>
> > "2e) How exactly is the '88% preservation to paired data performance' computed—unweighted average of ratios, or weighted by sample count? It feels a bit uneasy to me to average over different metrics/modalities to form this number."
>
> We acknowledge the notation could be clearer. We have now clarified throughout the paper:
>
> **Performance Preservation Ratio (PPR)** = (Average TextME score across variants / Paired upper-bound) × 100%
>
> **Example Calculation (AudioCaps R@1):**
> - TextME variants: Ours_CLIP (15.91), Ours_LB (14.54), Ours_NV (16.20), Ours_Qwen (15.35)
> - Average TextME score: (15.91 + 14.54 + 16.20 + 15.35) / 4 = 15.50
> - Paired upper-bound (Ours_upper-bound): 19.79
> - PPR = 15.50 / 19.79 × 100% = 78.3%
>
> **Aggregate Computation:**
> We compute PPR for each metric across both tables:
>
> | Table | Metrics | PPR Values |
> |-------|---------|------------|
> | **Retrieval** | AudioCaps R@1, R@5 | 78.3%, 83.6% |
> | | Clotho R@1, R@5 | 76.3%, 87.1% |
> | | DrugBank MRR@10, MRR@20 | 104.0%, 105.1% |
> | **Classification** | AudioSet mAP | 87.5% |
> | | ESC-50 Top-1, Top-5 | 112.7%, 102.6% |
> | | ModelNet40 Top-1, Top-5 | 93.6%, 97.8% |
> | | ScanObjectNN Top-1, Top-5 | 78.1%, 91.5% |
> | | RSNA Top-1 | 82.9% |
>
> **Overall Average:** Unweighted mean across all 14 task PPRs = **91.5%**
>
> This is now explicitly stated in Section 4.2.

---

### Author Response · Authors · 2025-11-21
**Official Comment to All Reviewers**

We sincerely thank all reviewers for the thoughtful and constructive feedback. Your insights have been invaluable in improving both the rigor and clarity of our work. Below, we summarize the key revisions addressing the shared concerns.

---

## Summary of Revisions

Based on the reviewers' feedback, we have made substantial revisions including:

1. **Theoretical justification** for modality gap hypotheses through InfoNCE analysis
2. **Clarified novelty and positioning** relative to prior modality gap and LLM-hub research
3. **Additional experiments**: offset ablation, cross-modal retrieval (3D→Image), pure text training validation, anchor space comparison, and image/video experiments
4. **Improved scientific writing** with appropriate qualifiers and a comprehensive Limitations section

---

## 1. Theoretical Validity of Hypotheses

**Concerns raised**: Reviewers questioned whether correlated noise, multi-cluster semantics, or nonlinear mappings could violate our assumptions.

**Our response**: We provide both theoretical and empirical justification.

- **Theorem 1 (Section 2.2)**: We prove that the InfoNCE objective intrinsically enforces noise-semantic orthogonality at optimum, preventing the correlated-noise scenario from occurring in contrastively-trained encoders.

- **Multi-cluster analysis**: While cluster-specific offsets show statistically significant differences, effect size analysis reveals that cluster membership explains less than 9% of offset variance, indicating bounded, predictable deviations consistent with our assumptions.

- **Nonlinearity tests**: Four independent statistical tests (CV, Spearman, distance correlation, MIC) confirm that nonlinear semantic dependencies are absent in naturally-paired encoders. We acknowledge CXR-CLIP text as a boundary case, likely due to synthetic caption generation.

- **Semantic preservation**: Fisher ratio analysis confirms that offset correction preserves class discriminability across all tested modalities.

---

## 2. Novelty and Positioning

**Concerns raised**: Reviewers noted overlap with prior modality gap research and LLM-hub approaches.

**Our clarification**: Our contribution is **not** gap analysis itself, but rather **exploiting** geometric properties for text-only modality expansion—a fundamentally different goal.

- **vs. Gap research**: Prior work discovers or mitigates gaps in paired-data settings. TextME is the first to exploit these properties for modality expansion using only text descriptions.

- **vs. LLM-hub approaches**: Methods like UniBind and LLM2CLIP require paired multimodal training. TextME enables expansion to specialized modalities lacking natural pairs, achieving comparable performance with significantly reduced data requirements.

We have revised the Introduction and Related Work sections with comprehensive positioning tables.

---

## 3. Experimental Completeness

**Offset ablation**: We demonstrate that offset correction is critical for most modalities, with substantial performance drops when removed. The key distinction from paired-data settings is that text-only training lacks implicit compensation mechanisms, making explicit geometric alignment essential.

**Pure text training**: Experiments with general-purpose corpora (Wiki1M, AllNLI) validate our text-only training claim, achieving meaningful cross-modal transfer without any modality-specific supervision.

**Additional experiments**: We have added 3D→Image retrieval, CLIP encoder ablation, and image/video retrieval results to demonstrate the framework's generalizability.

**Caption distribution analysis**: We provide systematic analysis explaining how domain specificity and train-test distribution shift affect performance across modalities.

---

## 4. Writing and Presentation

We have addressed presentation concerns including:

- Corrected Figure 1 interpretation (statistical independence, not clustering)
- Clarified notation and metric definitions throughout
- Toned down claims with appropriate qualifiers
- Added comprehensive Limitations section acknowledging dependencies on pretrained encoders, modality-specific offset effects, and domain-specific caption requirements

We are grateful for the opportunity to improve our work through this review process. All changes are highlighted in the revised manuscript, and we hope our revisions adequately address the reviewers' concerns. We welcome any further feedback.

---

### Meta-Review · Area_Chair_pVNy · 2026-01-05

**Summary:**

The paper develops an approach for multimodal learning expanding off of text as a kernel. The work builds off of the modality gap and computes a constant offset for each modality and trains a projection networks for sample efficiency. All the reviewers were negative initially with novelty, assumptions needed, and experiments as the main overlapping concerns. Some of these were addressed, but this paper is not sharp about the point it's trying to make, so I don't recommend acceptance.

**Reviewer Concerns:**

UG7e: Existence of of context translation vector. Text that is flashy and unqualified

MZaU: Novelty of the paper given the modality gap. The use of LLMs are the kernel or hub is already done. Questions about text/images

1UQr: Flaws in the theoretical construction, experiments, and visualizations

vP1a: Novelty, the use of text data that is still modality specific (text portion of existing multi modal datasets)

**Reviewer Scores:**

Given, the consensus I don't think any reviewers would flip a score to accept. Though I could see the 2's moving to a 4.

---

### Decision · Program_Chairs · 2026-01-26

Reject